# DECISION BOUNDARY VARIABILITY AND GENERALIZATION IN NEURAL NETWORKS

## ABSTRACT

This paper discovers that the neural network with lower decision boundary (DB) variability has better generalizability. Two new notions, *algorithm DB variability* and $(\epsilon, \eta)$-*data DB variability*, are proposed to measure the decision boundary variability from the algorithm and data perspectives. Extensive experiments show significant negative correlations between the decision boundary variability and the generalizability. From the theoretical view, two lower bounds based on algorithm DB variability are proposed and do not explicity depend on the sample size. By Chebyshev's inequality, we also prove two upper bounds of $\mathcal{O}\left(\sqrt{AV(f_{\mathbb{Q}}, \mathcal{D})/m(k-1)}\right)$ and $\mathcal{O}\left(\frac{1}{\sqrt{m}} + \epsilon + \eta \log \frac{1}{\eta}\right)$ based on algorithm and data DB variability, respectively. The algorithm DB variability upper bound is easier to calculate in practice, while the data DB variability upper bound relies on less assumptions. Moreover, the bounds do not explicitly depend on the network size, which is usually prohibitively large in deep learning.

## 1 INTRODUCTION

Neural networks (NNs) have achieved significant success in vast applications (Krizhevsky et al., 2012; Vaswani et al., 2017). However, the advance of NNs is arduous to be characterized by canonical statistical hypothesis complexity, such as VC-dimension (Vapnik et al., 1994) or Rademacher complexity (Bartlett & Mendelson, 2002). Considering the intimate connection between low variance and significant performance in learning theory, we investigate decision boundaries (DBs) from the perspective of variability in this paper. For neural networks, the decision boundary variability is largely caused by (1) the randomness from the training algorithm, and (2) the training data. In this paper, they are measured by two new terms, *algorithm DB variability* and $(\epsilon, \eta)$-*data DB variability*, and networks with lower DB variability are proved to possess better generalization performance.

Algorithm DB variability measures the variability of DBs in different training repeats, and extensive experiments are conducted on CIFAR-10, and CIFAR-100 (Krizhevsky & Hinton, 2009) datasets to explore the factors influence algorithm DB variability. We visualize the trend of the algorithm DB variability with respect to (*w.r.t.*) different training strategies, training time, sample sizes, and label noise. The empirical results present the negative correlation between algorithm DB variability and the test accuracy in all scenarios, which suggests that the algorithm DB variability largely indicates the generalization of neural networks. From the theoretical view, two lower bounds and an upper bound of the generalization error are proved based on the algorithm DB variability. The upper bound has a rate of $\mathcal{O}\left(\sqrt{AV(f_{\mathbb{Q}}, \mathcal{D})/m(k-1)}\right)$, where $AV(f_{\mathbb{Q}}, \mathcal{D})$ is the algorithm DB variability for $f_{\mathbb{Q}}$ on data generating distribution $\mathcal{D}$, $m$ is the sample size, and $k$ is the number of classes.

To present how the training data influences the decision boundary variability, the $(\epsilon, \eta)$-data DB variability is introduced by employing the $\eta$-subset to reconstruct decision boundaries with error $\epsilon$ (see Definition 5.1). If a decision boundary can be reconstructed by training on a smaller $\eta$-subset with a smaller reconstruction error $\epsilon$, the decision boundary possesses smaller data DB variability. Moreover, an $\eta$-$\epsilon$ curve can be drawn by varying the value of $\eta$ and numerically evaluating the label mismatch rate. The area under the $\eta$-$\epsilon$ curve could be a more elaborate indicator for the generalization of NNs because the curve contains richer information than the algorithm DB variability. An

$\mathcal{O}\left(\frac{1}{\sqrt{m}} + \epsilon + \eta \log \frac{1}{\eta}\right)$ generalization bound based on the $(\epsilon, \eta)$-data DB variability is established theoretically to enhance the relationship between the generalization of NNs and DB variability.

Compared to the algorithm DB variability upper bound, the data DB variability bound does not depend on additional Assumption 2, while the algorithm DB variability is easier to calculate than data DB variability in practice. Moreover, in contrast to many existing generalization bounds based on hypothesis complexity (Bartlett et al., 2019; Golowich et al., 2018; Bartlett et al., 2017) that require access to the network weight, our generalization bounds based on DB variability only demand the network predictions and thus have advantages in empirically approximating the generalization bound in (1) black-model settings, where model parameters are unavailable; and (2) over-parameterized settings, where calculating the weight norm is of prohibitively high computing burden.

## 2 RELATED WORKS

**Deep learning theory.** Due to the bias-variance trade-off, model complexity faces a dilemma in conventional wisdom (Mohri et al., 2018). Recently, Zhang et al. (2021) reveal the surprising ability of neural networks in fitting noise, but the networks still have impressive generalization performance in practice. This gap between the well-known bias-variance trade-off and the universal approximation ability for NNs draws attention to numerous researchers (Belkin et al., 2019; Nakkiran et al., 2019). Many works attribute the success of NNs to the effectiveness of the stochastic gradient descent (SGD) algorithm (Bottou, 2010; Hardt et al., 2016; Jin et al., 2017). Some empirical studies also explain the decent performance of networks by uncovering their learning properties (Nakkiran et al., 2020; Jiang et al., 2021; He et al., 2021). For instance, neural networks tend to fit the low-frequency information first (Rahaman et al., 2019; Xu et al., 2019) and gradually learn a more complex function (Kalimeris et al., 2019) during the training procedure.

**Decision boundaries and generalization.** Recent studies attempt to understand neural networks from the aspect of decision boundaries (He et al., 2018; Karimi et al., 2019). Guan & Loew (2020) empirically show the negative correlation between the complexity of decision boundary and generalization performance. Mickisch et al. (2020) reveal the phenomenon that the distance from data to decision boundaries continuously decreases during the training procedure. More recently, researchers find that NNs only rely on the most discriminative or simplest features to construct the DBs (Ortiz-Jimenez et al., 2020; Shah et al., 2020). Instead, our approach is different from these former methods by considering decision boundary variability, which is empirically and theoretically shown to closely correlate with the generalization in neural networks.

**Adversarial training and generalization.** It has been shown that the adversarial examples, which are created by adding non-perceivable perturbation on the input data, can completely mislead the NNs (Szegedy et al., 2013; Goodfellow et al., 2014). To tackle this problem, adversarial training is proposed to improve the robustness of the NNs through continuous training on adversarial examples (Madry et al., 2017). Nevertheless, Su et al. (2018) and Zhang et al. (2019) show a trade-off between the robustness and the generalization performance of NNs.

## 3 PRELIMINARIES

We denote the training set by $\mathcal{S} = \{(\mathbf{x}_i, y_i), i = 1, \ldots, m\}$, where $\mathbf{x}_i \in \mathbb{R}^n$, $n$ is the dimension of input data, $y_i \in \{1, \ldots, k\}$, $k$ is the number of classes, and $m = |\mathcal{S}|$ is the training sample size. We assume that $(\mathbf{x}_i, y_i)$ are independent and identically distributed (i.i.d.) random variables drawn from the data generating distribution $\mathcal{D}$. Denote the classifier as $f_{\boldsymbol{\theta}}(\mathbf{x}) : \mathbb{R}^n \to \mathbb{R}^k$, which is a neural network parameterized by $\boldsymbol{\theta}$. The output of $f_{\boldsymbol{\theta}}(\mathbf{x})$ is a $k$-dimensional vector and is assumed to be a discrete probability density function. Let $f_{\boldsymbol{\theta}}^{(i)}(\mathbf{x})$ be the $i$-th component of $f_{\boldsymbol{\theta}}(\mathbf{x})$, hence $\sum_{i=1}^{k} f_{\boldsymbol{\theta}}^{(i)}(\mathbf{x}) = 1$. We define $T(f_{\boldsymbol{\theta}}, \mathbf{x}) = \{i \in \{1, \cdots, k\} | f_{\boldsymbol{\theta}}^{(i)}(\mathbf{x}) = \max_j f_{\boldsymbol{\theta}}^{(j)}(\mathbf{x})\}$ to denote the set of predicted labels by $f_{\boldsymbol{\theta}}$ on $\mathbf{x}$. Due to the randomness of the learning algorithm $\mathcal{A}$, let $\mathbb{Q}(\boldsymbol{\theta}) = \mathcal{A}(\mathcal{S})$ denote the posteriori distribution returned by the learning algorithm $\mathcal{A}$ leveraged on the training set $\mathcal{S}$. Hence, we focus on the *Gibbs classifier* (a.k.a. random classifier) $f_{\mathbb{Q}} = \{f_{\boldsymbol{\theta}} | \boldsymbol{\theta} \sim \mathbb{Q}\}$. $0-1$ loss is employed in this paper, and the expected risks in terms of $\boldsymbol{\theta}$ and $\mathbb{Q}$ are defined as:

$$\mathcal{R}_{\mathcal{D}}(\boldsymbol{\theta}) = \mathbb{E}_{(\mathbf{x}, y) \sim \mathcal{D}} \left[\mathbb{I}\left(y \notin T\left(f_{\boldsymbol{\theta}}, \mathbf{x}\right)\right)\right] \quad (1)$$

and

$$\mathcal{R}_{\mathcal{D}}(\mathbb{Q}) = \mathbb{E}_{(\mathbf{x},y)\sim\mathcal{D}}\mathbb{E}_{\boldsymbol{\theta}\sim\mathbb{Q}}\left[\mathbb{I}\left(y \notin T\left(f_{\boldsymbol{\theta}},\mathbf{x}\right)\right)\right], \tag{2}$$

respectively. Here $\mathbb{I}(\cdot)$ is the indicator function. Since the data generating distribution $\mathcal{D}$ is unknown, evaluating the expected risk $\mathcal{R}_{\mathcal{D}}$ is not practical. Therefore, it is a practical way to estimate the expected risk by the empirical risk $\mathcal{R}_{\mathcal{S}}$, which is defined as:

$$\mathcal{R}_{\mathcal{S}}(\boldsymbol{\theta}) = \mathbb{E}_{(\mathbf{x},y)\sim\mathcal{S}}\left[\mathbb{I}\left(y \notin T\left(f_{\boldsymbol{\theta}},\mathbf{x}\right)\right)\right] = \frac{1}{m}\sum_{i=1}^{m}\mathbb{I}\left(y_i \notin T\left(f_{\boldsymbol{\theta}},\mathbf{x}_i\right)\right) \tag{3}$$

$$\mathcal{R}_{\mathcal{S}}(\mathbb{Q}) = \mathbb{E}_{(\mathbf{x},y)\sim\mathcal{S}}\mathbb{E}_{\boldsymbol{\theta}\sim\mathbb{Q}}\left[\mathbb{I}\left(y \notin T\left(f_{\boldsymbol{\theta}},\mathbf{x}\right)\right)\right] = \frac{1}{m}\sum_{i=1}^{m}\mathbb{E}_{\boldsymbol{\theta}\sim\mathbb{Q}}\left[\mathbb{I}\left(y_i \notin T\left(f_{\boldsymbol{\theta}},\mathbf{x}_i\right)\right)\right], \tag{4}$$

where $(\mathbf{x}_i, y_i) \in \mathcal{S}$ and $m = |\mathcal{S}|$.

## 3.1 DECISION BOUNDARY

If the output $k$-dimensional vector $f_{\boldsymbol{\theta}}(\mathbf{x})$ on the input example $\mathbf{x}$ has a tie, *i.e.*, the maximum value of the vector is not unique, then $\mathbf{x}$ is considered to locate on the decision boundary of $f_{\boldsymbol{\theta}}$. With this idea, the decision boundary can be formally defined as below:

**Definition 3.1** (decision boundary). Let $f_{\boldsymbol{\theta}}(\mathbf{x}) : \mathbb{R}^n \to \mathbb{R}^k$ be a classifier network parameterized by $\boldsymbol{\theta}$. Then the *decision boundary* of $f_{\boldsymbol{\theta}}$ is defined by

$$\{\mathbf{x} \in \mathbb{R}^n | \exists i, j \in \{1, \cdots, k\}, i \neq j, f_{\boldsymbol{\theta}}^{(i)}(\mathbf{x}) = f_{\boldsymbol{\theta}}^{(j)}(\mathbf{x}) = \max_q f_{\boldsymbol{\theta}}^{(q)}(\mathbf{x})\} \tag{5}$$

After defining the decision boundary, we have the following remark:

**Remark 1.** (1) If an input example $(\mathbf{x}, y)$ is not located on the decision boundary of $f_{\boldsymbol{\theta}}$, $T(f_{\boldsymbol{\theta}}, \mathbf{x})$ is a singleton, and we have

$$\mathbb{I}\left(y \notin T\left(f_{\boldsymbol{\theta}},\mathbf{x}\right)\right) = \mathbb{I}\left(y \neq T\left(f_{\boldsymbol{\theta}},\mathbf{x}\right)\right). \tag{6}$$

(2) If the input $\mathbf{x}$ is a boundary point, in practice, we randomly draw a label from the set $T(f_{\boldsymbol{\theta}}, \mathbf{x})$ as the prediction of $f_{\boldsymbol{\theta}}$ on $\mathbf{x}$.

## 3.2 ADVERSARIAL TRAINING

Adversarial training (AT) is a popular strategy to enhance the adversarial robustness of neural networks against adversarial examples, which is generated through projected gradient descent (PGD) (Madry et al., 2018) in our empirical studies. More specifically, adversarial training can be formulated as solving the minimax-loss problem as follows:

$$\min_{\boldsymbol{\theta}} \frac{1}{m} \sum_{i=1}^{m} \max_{\|\mathbf{x}_i' - \mathbf{x}_i\| \leq \gamma} \ell\left(f_{\boldsymbol{\theta}}\left(\mathbf{x}_i'\right), y_i\right), \tag{7}$$

where $\gamma$ is the *radius* to limit the distance between adversarial examples and original examples. Adversarial training can be considered to enlarge the minimum distances from training examples to decision boundaries to at least $\gamma$.

## 4 ALGORITHM DECISION BOUNDARY VARIABILITY

Due to the randomness of learning algorithms, there is no doubt that the parameters have a substantial variation when the network is repeatedly trained. However, *do the decision boundaries in these repeated-training networks have a large discrepancy?* With this question, we define the algorithm decision boundary variability (AV) to measure the variability of DBs caused by the randomness of algorithms in different training repeats.

**Definition 4.1** (algorithm decision boundary variability). Let $f_{\boldsymbol{\theta}}(\mathbf{x}) : \mathbb{R}^n \to \mathbb{R}^k$ be a classifier network parameterized by $\boldsymbol{\theta}$ and let $\mathbb{Q}(\boldsymbol{\theta})$ be the distribution over $\boldsymbol{\theta}$. Then, the algorithm decision boundary variability for $f_{\mathbb{Q}}$ on $\mathcal{D}$ is

$$AV(f_{\mathbb{Q}}, \mathcal{D}) = \mathbb{E}_{(\mathbf{x},y)\sim\mathcal{D}}\mathbb{E}_{\boldsymbol{\theta},\boldsymbol{\theta}'\sim\mathbb{Q}}\left[\mathbb{I}(T(f_{\boldsymbol{\theta}},\mathbf{x}) \neq T(f_{\boldsymbol{\theta}'},\mathbf{x}))\right], \tag{8}$$

where $T(f_{\boldsymbol{\theta}},\mathbf{x}) = \{i \in \{1, \cdots, k\}|f_{\boldsymbol{\theta}}^{(i)}(\mathbf{x}) = \max_j f_{\boldsymbol{\theta}}^{(j)}(\mathbf{x})\}$.

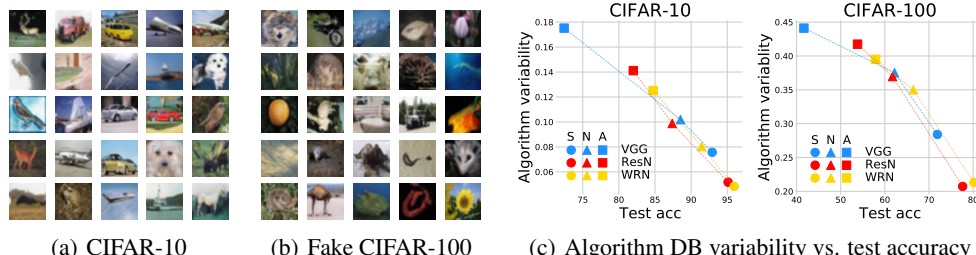

(a) CIFAR-10      (b) Fake CIFAR-100      (c) Algorithm DB variability vs. test accuracy

Figure 1: Algorithm decision boundary variability on CIFAR-10 and CIFAR-100. (a) Examples of fake CIFAR-10 images generated by conditional BigGAN. (b) Examples of fake CIFAR-100 images generated by conditional BigGAN. (c) Scatter plots of algorithm DB variability to accuracy on test set with different architectures and training strategies on CIFAR-10 and CIFAR-100. The colors of blue, red, and yellow points denote the architectures of VGG-16 (VGG), ResNet-18 (ResN), and WideResNet-28 (WRN), respectively. The shapes of ●, ▲, and ■ designate the training strategies of standard training (S), non-data-augmentation training (N), and adversarial training (A), respectively. Each point is calculated and then averaged on 10 trials.

In the light of Definition 4.1, smaller algorithm DB variability $AV(f_{\mathbb{Q}}, \mathcal{D})$ means that the network produces more stable DBs or converges to more similar DBs during different training repeats. The algorithm DB variability is also a good notion for the "entropy" of decision boundaries because it measures the decision uncertainty over the whole data generating distribution; we provide a detailed discussion in Appendix B.1 due to space limitation.

## 4.1 ALGORITHM DB VARIABILITY AND GENERALIZATION

To explore the relationship between the algorithm DB variability and generalization in neural networks, we conduct experiments with different network architectures and training strategies on CIFAR-10 and CIFAR-100. In detail, VGG-16 (Simonyan & Zisserman, 2014), ResNet-18 (He et al., 2016), and WideResNet-28 (Zagoruyko & Komodakis, 2016b) are optimized by standard, non-data-augmentation and adversarial training, respectively, until the training procedure converges. We clarify that basic data augmentation (crop and flip) (Zagoruyko & Komodakis, 2016a) is adopted in both standard and adversarial training, and only the basic data augmentation is considered in our experiments and analysis. Each training setting (dataset, architecture, training strategy) is repeated for 10 trials with different random seeds to estimate the parameter distribution $\mathbb{Q}(\boldsymbol{\theta})$. As for simulating the data generating distribution, we trained two conditional BigGAN (Zhao et al., 2020), an advanced generative network architecture, to produce $100,000$ fake images for CIFAR-10 and CIFAR-100, respectively. Examples of fake images have been shown in Figure 1(a) and 1(b).

Through these generative fake images, we can estimate the algorithm DB variability *w.r.t.* these well-trained models. For each training setting, we plot both the average test accuracy and algorithm DB variability; see Figure 1(c). From the plots, we have several observations: (1) adversarial training dramatically decreases the test accuracy and promotes the algorithm DB variability compared to standard training. (2) data augmentation decreases the algorithm decision boundary variability by comparing the standard training and non-data-augmentation training. The reason is that the augmented images by cropping or flipping are still located on the data generating distribution, so data augmentation is considered to expand the size of the training set. Hence, the expanded training set can characterize longer or larger decision boundaries on the data generating distribution; (3) WideResNet has better test accuracy and lower algorithm DB variability than ResNet and VGG; and (4) there is a negative correlation between the test accuracy and the algorithm DB variability. Therefore, based on these findings, we propose the following conjecture:

**Hypothesis 1.** *Neural networks with smaller algorithm decision boundary variability on the data generating distributions possess better generalization performance.*

We then conduct experiments on the algorithm DB variability *w.r.t.* training time, sample size, and label noise to concrete this hypothesis.

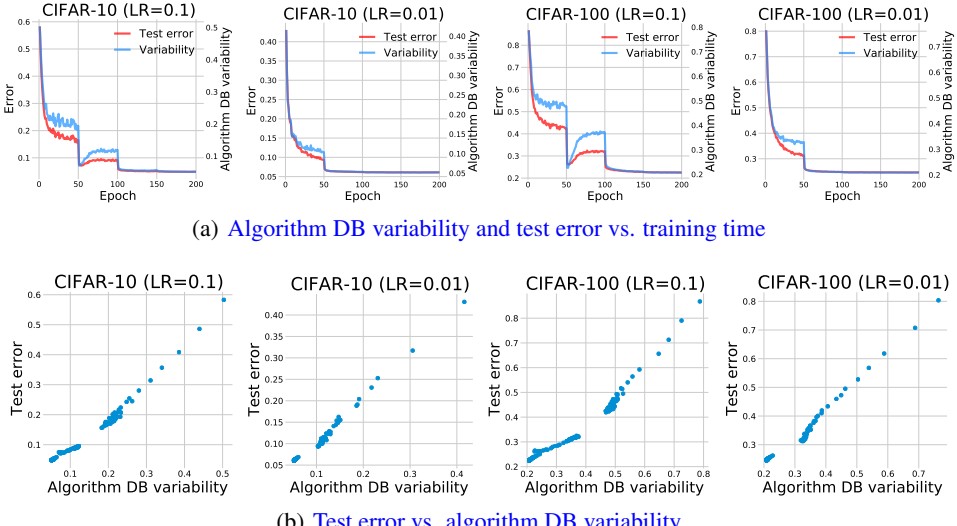

(a) Algorithm DB variability and test error vs. training time

(b) Test error vs. algorithm DB variability

Figure 2: (a) Plots of algorithm DB variability and test error as functions of training time (LR is learning rate). (b) Scatter plots of test error to algorithm DB variability (LR is learning rate). The points are collected from different epochs. Each curve and point is calculated and then averaged on 10 trials.

## 4.2 ALGORITHM DB VARIABILITY AND TRAINING TIME

To investigate the relationship between algorithm DB variability and training time, we train 40 ResNet-18 with different initial learning rates of 0.1 and 0.01 on CIFAR-10 and CIFAR-100. Then, the algorithm DB variability and test error are calculated at each epoch; see Figure 2(a). From the plots, we can observe that (1) algorithm DB variability and test error share a very similar curve *w.r.t.* the training time; and (2) algorithm DB variability decreases during the training process. The decline of algorithm DB variability shows that the interpolation on examples reduces the variability of decision boundaries on data generating distribution. As shown in Figure 2(b), we collect the points of (algorithm DB variability, test error) from different epochs, and the scatter plots present a significant positive correlation between test error and the algorithm DB variability, and thus supports Hypothesis 1.

## 4.3 ALGORITHM DB VARIABILITY AND SAMPLE SIZE

We next investigate how sample size impacts the algorithm DB variability. 100 ResNet-18 are trained on five training sample sets of different sizes randomly drawn from CIFAR-10 and CIFAR-100, while all irrelevant variables are strictly controlled. Then, the algorithm DB variability and test error are calculated in all cases; see Figure 3(a). From the plots, we have the following observations: (1) test error and algorithm DB variability share a very similar curve *w.r.t.* the training sample size; (2) larger sample size, which intuitively helps obtain a more smooth estimation of the decision boundary, also contributes to smaller algorithm DB variability; and (3) there is a significant positive correlation between test error and algorithm DB variability, which fully supports Hypothesis 1.

## 4.4 ALGORITHM DB VARIABILITY AND LABEL NOISE

Belkin et al. (2019); Nakkiran et al. (2019) show the surprising epoch-wise double descent of test error, especially with the existence of label noise. In this section, we explore the trend of algorithm DB variability when the label noise exists. We train 20 ResNet-10 for 500 epoch with a constant learning rate of 0.0001 on CIFAR-10 and CIFAR-100 with 20% label noise. We clarify that the noise labels remain unchanged during different training repeats, which is necessary to estimate the algorithm DB variability. Then, the average test error and algorithm DB variability are calculated at each training epoch, as shown in Figure 3(b). From the plots, we observe that: (1) test error and

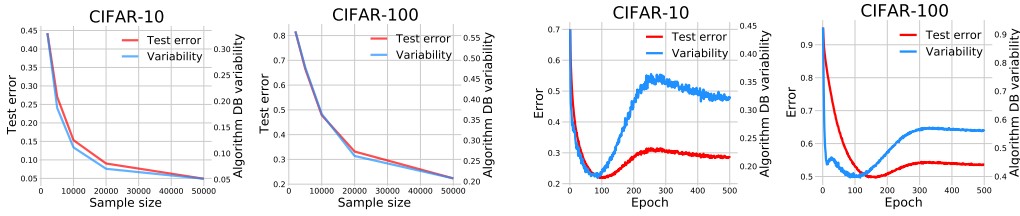

(a) Algorithm DB variability vs. sample size          (b) Algorithm DB variability vs. time (label noise)

Figure 3: (a) Plots of algorithm DB variability and test error as functions of training sample size on CIFAR-10 and CIFAR-100. (b) Plots of algorithm DB variability and test error as functions of training time with the existence of 20% label noise on CIFAR-10 and CIFAR-100. Each curve is calculated and then averaged on 10 trials.

algorithm DB variability still share a very similar curve *w.r.t.* the training time with the existence of label noise; and (2) the algorithm DB variability also undergoes an epoch-wise double descent during the training process, especially in the left panel of Figure 3(b), which implies that factors influence the generalization of networks can also have an influence on the algorithm DB variability. Hence, algorithm DB variability is an excellent indicator for the generalization ability of networks.

## 4.5 THEORETICAL EVIDENCE

In this section, we explore and develop the theoretical foundations for the algorithm decision boundary variability on data generating distributions.

**Assumption 1.** The decision boundary of the classifier network $f_{\boldsymbol{\theta}}$ on data generation distribution $\mathcal{D}$ is a set with measure zero.

**Corollary 1.** Let $f_{\boldsymbol{\theta}}(\mathbf{x}) : \mathbb{R}^n \to \mathbb{R}^k$ be a classifier network parameterized by $\boldsymbol{\theta}$. If Assumption 1 holds for all $\boldsymbol{\theta} \sim \mathbb{Q}$, then, for all $i \in \{1, \cdots, k\}$, we have

$$\mathbb{E}_{(\mathbf{x},y)\sim\mathcal{D}}\left[\mathbb{I}(i \in T(f_{\boldsymbol{\theta}}, \mathbf{x}))\right] = \mathbb{E}_{(\mathbf{x},y)\sim\mathcal{D}}\left[\mathbb{I}(T(f_{\boldsymbol{\theta}}, \mathbf{x}) = i)\right] \tag{9}$$

and

$$\mathbb{E}_{(\mathbf{x},y)\sim\mathcal{D}}\left[\mathbb{I}(i \notin T(f_{\boldsymbol{\theta}}, \mathbf{x}))\right] = \mathbb{E}_{(\mathbf{x},y)\sim\mathcal{D}}\left[\mathbb{I}(T(f_{\boldsymbol{\theta}}, \mathbf{x}) \neq i)\right]. \tag{10}$$

### 4.5.1 ALGORITHM DB VARIABILITY-BASED LOWER BOUNDS

**Theorem 1** (lower bound on expected risk). Let $f_{\boldsymbol{\theta}}(\mathbf{x}) : \mathbb{R}^n \to \mathbb{R}^k$ be a classifier network parameterized by $\boldsymbol{\theta}$ and let $\mathbb{Q}(\boldsymbol{\theta})$ be the distribution over $\boldsymbol{\theta}$. Then, if Assumption 1 holds for all $\boldsymbol{\theta} \sim \mathbb{Q}$, we have

$$\mathcal{R}_{\mathcal{D}}(\mathbb{Q}) \geq \frac{AV(f_{\mathbb{Q}}, \mathcal{D})}{2}, \tag{11}$$

where $AV(f_{\mathbb{Q}}, \mathcal{D}) = \mathbb{E}_{(\mathbf{x},y)\sim\mathcal{D}}\mathbb{E}_{\boldsymbol{\theta},\boldsymbol{\theta}'\sim\mathbb{Q}}\left[\mathbb{I}\left(T(f_{\boldsymbol{\theta}}, \mathbf{x}) \neq T(f_{\boldsymbol{\theta}'}, \mathbf{x})\right)\right]$ is the algorithm DB variability for $f_{\mathbb{Q}}$ on data generating distribution $\mathcal{D}$.

The proof is given in Appendix C.1. Theorem 1 provides a lower bound on the expected risk $\mathcal{R}_{\mathcal{D}}(\mathbb{Q})$ based on the algorithm DB variability $AV(f_{\mathbb{Q}}, \mathcal{D})$, and the lower bound presents that the Gibbs classifier $f_{\mathbb{Q}}$ possesses a significant expected risk when its algorithm DB variability is large. Moreover, when we consider the binary classification, *i.e.*, $k = 2$, there is a tighter lower bound:

**Theorem 2** (lower bound for binary case). Let $f_{\boldsymbol{\theta}}(\mathbf{x}) : \mathbb{R}^n \to \mathbb{R}^2$ be a binary classifier network parameterized by $\boldsymbol{\theta}$ and let $\mathbb{Q}(\boldsymbol{\theta})$ be the distribution over $\boldsymbol{\theta}$. Suppose the expected risk $\mathcal{R}_{\mathcal{D}}(\mathbb{Q}) \leq \frac{1}{2}$ and Assumption 1 hold for all $\boldsymbol{\theta} \sim \mathbb{Q}$, then we have

$$\mathcal{R}_{\mathcal{D}}(\mathbb{Q}) \geq \frac{1 - \sqrt{1 - 2AV(f_{\mathbb{Q}}, \mathcal{D})}}{2}. \tag{12}$$

### 4.5.2 An algorithm DB variability-based upper bound

Before we present the upper bound based on the algorithm DB variability, let us introduce the assumption used in this section.

**Assumption 2.**

$$\mathbb{E}_{(\mathbf{x},y)\sim\mathcal{D}}\mathbb{E}_{\boldsymbol{\theta}\sim\mathbb{Q}}^2\left[\mathbb{I}\left(y\notin T\left(f_{\boldsymbol{\theta}},\mathbf{x}\right)\right)\right]\leq\frac{2}{k+1}\mathcal{R}_{\mathcal{D}}(\mathbb{Q}),\tag{13}$$

where $i\in\{1,2,\cdots,k\}$ and $k$ is the number of potential categories.

With Assumption 2, we can derive the upper bound on the variance of risk:

**Lemma 1** (upper bound on risk variance). Let $f_{\boldsymbol{\theta}}(\mathbf{x}):\mathbb{R}^n\to\mathbb{R}^k$ be a classifier network parameterized by $\boldsymbol{\theta}$ and let $\mathbb{Q}(\boldsymbol{\theta})$ be the distribution over $\boldsymbol{\theta}$. Suppose Assumption 1 and 2 hold, then we have

$$\mathrm{Var}_{(\mathbf{x},y)\sim\mathcal{D}}\left[\mathbb{E}_{\boldsymbol{\theta}\sim\mathbb{Q}}\left[\mathbb{I}\left(y\notin T\left(f_{\boldsymbol{\theta}},\mathbf{x}\right)\right)\right]\right]\leq\frac{AV(f_{\mathbb{Q}},\mathcal{D})}{k-1},\tag{14}$$

where $AV(f_{\mathbb{Q}},\mathcal{D})=\mathbb{E}_{(\mathbf{x},y)\sim\mathcal{D}}\mathbb{E}_{\boldsymbol{\theta},\boldsymbol{\theta}'\sim\mathbb{Q}}\left[\mathbb{I}\left(T\left(f_{\boldsymbol{\theta}},\mathbf{x}\right)\neq T\left(f_{\boldsymbol{\theta}'},\mathbf{x}\right)\right)\right]$ is the algorithm DB variability of $f_{\mathbb{Q}}$ on data generating distribution $\mathcal{D}$.

The proof is given in Appendix C.3. Hence, the next theorem is a direct consequence of the *one-sided Chebyshev's inequality*: $\mathrm{Pr}\left[\mathbb{E}[Z_1]-\frac{1}{m}\sum_{i=1}^m Z_i>a\right]\leq\frac{\mathrm{Var}[Z_1]}{2ma^2}$ for any $a>0$.

**Theorem 3** (PAC upper bound on expected risk). Let $f_{\boldsymbol{\theta}}(\mathbf{x}):\mathbb{R}^n\to\mathbb{R}^k$ be a classifier network parameterized by $\boldsymbol{\theta}$ and let $\mathbb{Q}(\boldsymbol{\theta})$ be the distribution over $\boldsymbol{\theta}$ *w.r.t.* the training set $\mathcal{S}$. Suppose Assumption 1 and 2 hold. Then, with the probability of at least $1-\delta$ over a sample of size $m$, we have

$$\mathcal{R}_{\mathcal{D}}(\mathbb{Q})\leq\mathcal{R}_S(\mathbb{Q})+\sqrt{\frac{AV(f_{\mathbb{Q}},\mathcal{D})}{2m(k-1)\delta}}.\tag{15}$$

The above theorem presents that a smaller algorithm DB variability $AV(f_{\mathbb{Q}},\mathcal{D})$ contributes to a tighter upper bound on the true risk, which theoretically verifies Hypothesis 1 that neural networks with smaller algorithm decision boundary variability possess better generalization performance.

## 5 Data decision boundary variability

In the previous sections, we introduced the algorithm DB variability, which measures the decision boundary variability caused by the randomness of learning algorithms. However, the algorithm DB variability hardly shows the decision boundary variability caused by changes in training data. To remedy the problem and complete our theorem about decision boundary variability, we define the data DB variability as below:

**Definition 5.1** (data decision boundary variability). Let $f_{\boldsymbol{\theta}}(\mathbf{x}):\mathbb{R}^n\to\mathbb{R}^k$ be a classifier network parameterized by $\boldsymbol{\theta}$, where $\boldsymbol{\theta}\sim\mathcal{A}(\mathcal{S})$ is returned by leveraging the stochastic learning algorithm $\mathcal{A}(\mathcal{S})$ on the training set $\mathcal{S}$, which is sampled from the data generating distribution $\mathcal{D}$. We term $\mathcal{S}_\eta\subset\mathcal{S}$ as a $\eta$-*subset* of $\mathcal{S}$ if $\frac{|\mathcal{S}_\eta|}{|\mathcal{S}|}=\eta$. Then, if we fixed $\eta$ and

$$\inf_{\mathcal{S}_\eta\subset\mathcal{S}}\mathbb{E}_{(\mathbf{x},y)\sim\mathcal{D}}\mathbb{E}_{\boldsymbol{\theta}\sim\mathcal{A}(\mathcal{S}),\boldsymbol{\theta}'\sim\mathcal{A}(\mathcal{S}_\eta)}\left[\mathbb{I}\left(T(f_{\boldsymbol{\theta}},\mathbf{x})\neq T(f_{\boldsymbol{\theta}'},\mathbf{x})\right)\right]=\epsilon,\tag{16}$$

the decision boundary of $f_{\mathcal{A}(\mathcal{S})}$ is said to possess a $(\epsilon,\eta)$-*data decision boundary variability*.

The data decision boundary variability contains two parameters of $\epsilon$ and $\eta$, respectively. That Gibbs classifier $f_{\mathcal{A}(\mathcal{S})}$ has a $(\epsilon,\eta)$-data DB variability means that only the proportion of $\eta$ of $\mathcal{S}$, *i.e.*, $\mathcal{S}_\eta$, which can be considered as "support vector set", is enough to reconstruct a similar decision boundary with the reconstruction error $\epsilon$. The data DB variability can also be connected with the complexity of decision boundaries if we assume that simpler decision boundaries rely on fewer "support vectors"; we provide a detailed discussion in Appendix B.2 due to space limitation.

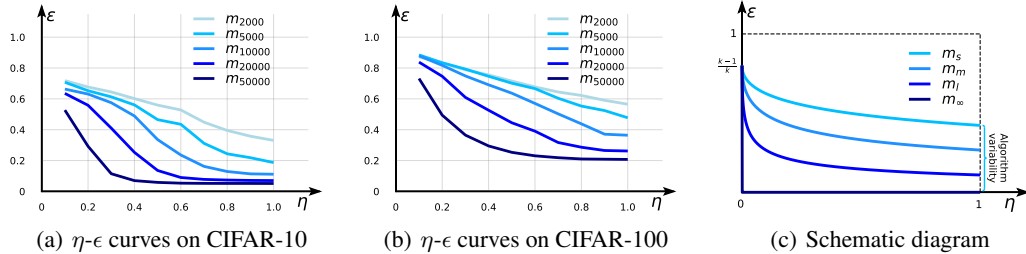

(a) $\eta$-$\epsilon$ curves on CIFAR-10     (b) $\eta$-$\epsilon$ curves on CIFAR-100     (c) Schematic diagram

Figure 4: (a) The $\eta$-$\epsilon$ curves on CIFAR-10 with different training sample sizes 2000 ($m_{2000}$), 2000 ($m_{2000}$), 10000 ($m_{10000}$), 20000 ($m_{20000}$), and 50000 ($m_{50000}$), respectively. (b) The $\eta$-$\epsilon$ curves on CIFAR-100 with different training sample sizes. (c) The schematic diagram of the $\eta$-$\epsilon$ curves *w.r.t.* small ($m_s$), medium ($m_m$), large ($m_l$), and infinite ($m_\infty$) sample sizes, respectively.

### 5.1   $\eta$-$\epsilon$ CURVES ABOUT DATA DB VARIABILITY

According to Definition 5.1, the data DB variability degrades to the algorithm DB variability $AV(f_{\mathbb{Q}}, \mathcal{D})$ when $\mathcal{S}_\eta = \mathcal{S}$. In other words, the algorithm DB variability is a special case of the data DB variability with $\eta = 1$. Therefore, the data DB variability could present more detailed information on reflecting how the decision boundary variability depends on the training set, especially when we observe the variation of the reconstruction error $\epsilon$ *w.r.t.* different $\eta$.

To explore the relationship between the reconstruction error $\epsilon$ and the proportion of subset $\eta$, we train 1000 networks of ResNet-18 on CIFAR-10 and CIFAR-100 of different sample sizes $m$, respectively. Albeit finding the most suitable $\eta$-subset is intractable, we adopt a coreset selection approach named *selection via proxy* (Coleman et al., 2020), which can rank the importance of training examples, to estimate the $\eta$-subset for a given training set $\mathcal{S}$ and proportion $\eta$ (full list of $\eta$ involved in our experiments is presented in Appendix A.3.6). Then, through repeatedly training the network on $\mathcal{S}_\eta$, we can estimate the reconstruction error $\epsilon$. The $\eta$-$\epsilon$ curves of CIFAR-10 and CIFAR-100 are presented in Figure 4(a) and 4(b), respectively. From the plots, we have an observation that *there is a more rapid decline in $\epsilon$ along with small $\eta$ and also a smaller algorithm DB variability when the training sample size $m$ is larger*. Furthermore, we plot the schematic diagram of $\eta$-$\epsilon$ curves *w.r.t.* different sample size $m$, as shown in Figure 4(c). When $\eta = 0$, $f_{\mathcal{A}(\mathcal{S}_n)}$ cannot be better than random guess, and hence $\epsilon = \frac{k-1}{k}$, where $k$ is the number of potential categories. We have noticed that $\epsilon$ has a sharper drop along with $\eta$ when the sample size $m$ is larger. Therefore, we rationally propose the following assumption, which is also shown by the right angle with $m_\infty$ in Figure 4(c).

**Assumption 3.** If $m \to \infty$, then we have $\epsilon \to 0$ when $\eta \to 0$.

These plots indicate that the area under the $\eta$-$\epsilon$ curve could be a more meticulous predictor for the generalization ability of neural networks compared to the algorithm DB variability, which is only a point on the $\eta$-$\epsilon$ curve when $\eta = 1$. Hence, the area under the $\eta$-$\epsilon$ curve can also be considered as an extension of the algorithm DB variability: if the Gibbs classifier $f_{\mathcal{A}(\mathcal{S})}$ possesses a smaller area under the $\eta$-$\epsilon$ curve, it produces more stable decision boundaries with varying training subsets. In the following section, we will theoretically establish the connection between the parameters of $\eta$ and $\epsilon$ and the generalization ability of neural networks.

### 5.2   THEORETICAL EVIDENCE

In this section, we develop the theoretical foundations for the data decision boundary variability, and present that *neural networks with better data DB variability possess better generalization*.

**Lemma 2.** If the decision boundaries of $f_{\mathcal{A}(\mathcal{S})}$ possess a $(\epsilon, \eta)$-data DB variability, then we have

$$|\mathcal{R}_{\mathcal{D}}(\mathcal{A}(\mathcal{S})) - \mathcal{R}_{\mathcal{D}}(\mathcal{A}(\mathcal{S}_\eta))| \leq \epsilon. \tag{17}$$

Lemma 2 shows that the difference between the expected risk of $\mathcal{A}(\mathcal{S})$ and $\mathcal{A}(\mathcal{S}_\eta)$ can be bounded by their difference in decision boundaries. Then, by applying the concentration inequality, we can

bound the difference between their empirical risk on the same training set $\mathcal{S}$ with the probability $1 - \delta$:

**Lemma 3.** If the decision boundaries of $f_{\mathcal{A}(\mathcal{S})}$ possess a $(\epsilon, \eta)$-data DB variability, then, with probability of at least $1 - \delta$ over a sample of size $m$, we have

$$\mathcal{R}_{\mathcal{S}}(\mathcal{A}(\mathcal{S}_\eta)) \leq \mathcal{R}_{\mathcal{S}}(\mathcal{A}(\mathcal{S})) + \sqrt{\frac{1}{2m} \log \frac{1}{\delta}} + \epsilon, \tag{18}$$

where $m = |\mathcal{S}|$ is the training sample size. If we further assume the training error $\mathcal{R}_{\mathcal{S}}(\mathcal{A}(\mathcal{S})) = 0$, then, with the probability of at least $1 - \delta$ over a sample of size $m$, we have

$$\mathcal{R}_{\mathcal{S}}(\mathcal{A}(\mathcal{S}_\eta)) \leq \sqrt{\frac{1}{2m} \log \frac{1}{\delta}} + \epsilon \tag{19}$$

**Theorem 4** (data DB variability-based upper bound on expected risk). If the decision boundaries of $f_{\mathcal{A}(\mathcal{S})}$ possess a $(\epsilon, \eta)$-data DB variability on the data generating distribution $\mathcal{D}$, and assume $\eta \leq 0.5$ and the empirical risks $\mathcal{R}_{\mathcal{S}}(\mathcal{A}(\mathcal{S})) = \mathcal{R}_{\mathcal{S}_\eta}(\mathcal{A}(\mathcal{S}_\eta)) = 0$, then, with the probability of at least $1 - \delta$ over a sample of size $m$, we have

$$\mathcal{R}_{\mathcal{D}}(\mathcal{A}(\mathcal{S})) \leq \Omega + \sqrt{4\Omega\Delta} + 8\Delta + \epsilon, \tag{20}$$

where

$$\Omega = \frac{1}{1 - \eta} \left( \sqrt{\frac{1}{2m} \log \frac{2}{\delta}} + \epsilon \right), \tag{21}$$

$$\Delta = \eta \log \frac{e}{\eta} + \frac{1}{m} \log \frac{2}{\delta}. \tag{22}$$

Moreover, for sufficient large $m$, we have

$$\mathcal{R}_{\mathcal{D}}(\mathcal{A}(\mathcal{S})) \leq \mathcal{O}(\frac{1}{\sqrt{m}} + \epsilon + \eta \log \frac{1}{\eta}). \tag{23}$$

The proof is omitted here and is given in Appendix C.7. According to Assumption 3, when $m \to \infty$, then $\eta \to 0$ and $\epsilon \to 0$, and according to Eq. 23, $\mathcal{R}_{\mathcal{D}}(\mathcal{A}(\mathcal{S})) \to 0$. Therefore, the generalization bound is asymptotically converged. Theorem 4 presents that smaller data DB variability, *i.e.*, smaller $\epsilon$ and $\eta$, contributes to a tighter upper bound on the expected risk, and also theoretically verifies the relationship between the data DB variability and the generalization ability of neural networks.

## 6 CONCLUSION AND DISCUSSION

In this paper, we empirically and theoretically explored the relationship between decision boundary variability and generalization in neural networks, through the algorithm DB variability and data DB variability, respectively. A significant negative correlation between the decision boundary variability and generalization performance is observed in our experiments. As for the theoretical results, two lower bounds and two upper bounds were proposed based on algorithm DB variability and data DB variability to enhance our findings. The algorithm DB variability upper bound is easier to calculate in practice, while the data DB variability upper bound relies on less assumptions. Moreover, in contrast to many existing generalization bounds based on hypothesis complexity that require access to the network weight, our generalization bounds based on decision boundary variability only demand access to the network predictions and thus have advantages in empirically approximating the generalization bound in (1) black-model settings, where model parameters are unavailable; and (2) over-parameterized settings, where calculating the weight norm is of prohibitively high computing burden. Furthermore, the analysis in Section 5.1 mentioned that the algorithm DB variability is a special case of the data DB variability. Therefore, unifying the theoretical works about algorithm DB variability and data DB variability will be a promising direction in future works.

## ACKNOWLEDGEMENT

The authors sincerely appreciate the anonymous reviewers for their valuable suggestions.

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

## A   ADDITIONAL EXPERIMENTS DETAILS

This section provides all the additional details of our experiments.

### A.1   DATASETS

Our experiments are conducted on two public datasets (CIFAR-10 and CIFAR-100 (Krizhevsky & Hinton, 2009)) and two manufactured datasets (fake CIFAR-10, and fake CIFAR-100). The detail of these datasets are shown as follows.

- **CIFAR-10** consists of $50,000$ training images and $10,000$ test images from 10 different classes, and **CIFAR-100** consists of $50,000$ training images and $10,000$ test images from 100 different classes. One can download CIFAR-10 and CIFAR-100 from `https://www.cs.toronto.edu/~kriz/cifar.html`.

- **Fake CIFAR-10** consists of $100,000$ test images from 10 different classes. The fake CIFAR-10 is generated by conditional BigGAN (Zhao et al., 2020). The pre-trained Big-GAN is provided by Kang & Park (2020) and can be download from `https://drive.google.com/drive/folders/1xVN7dQPWMLi8gDZEb5FThkjbFtIdzb6b`.

- **Fake CIFAR-100** consists of $100,000$ test images from 100 different classes. The generation of fake CIFAR-100 is similar to it of fake CIFAR-10, while the training set for the BigGAN is replaced from CIFAR-10 to CIFAR-100.

### A.2   MODEL ARCHITECTURES

We use different neural network architectures in our experiments, including VGG-16, ResNet-18, and WideResNet-28. The architectures are presented in Table 1.

Table 1: Detailed model architectures for CIFAR-10/100

| VGG-16 | ResNet-18 | WideResNet-28-10 |
|---|---|---|
| $(3 \times 3, 32) \times 2$ 
 maxpool, $2 \times 2$ | $3 \times 3, 64$ | $3 \times 3, 16$ |
| $(3 \times 3, 128) \times 2$ 
 maxpool, $2 \times 2$ | $\begin{bmatrix} 3 \times 3, 64 \\ 3 \times 3, 64 \end{bmatrix} \times 2$ | $\begin{bmatrix} 3 \times 3, 160 \\ 3 \times 3, 160 \end{bmatrix} \times 4$ |
| $(3 \times 3, 256) \times 3$ 
 maxpool, $2 \times 2$ | $\begin{bmatrix} 3 \times 3, 128 \\ 3 \times 3, 128 \end{bmatrix} \times 2$ | $\begin{bmatrix} 3 \times 3, 320 \\ 3 \times 3, 320 \end{bmatrix} \times 4$ |
| $(3 \times 3, 512) \times 3$ 
 maxpool, $2 \times 2$ | $\begin{bmatrix} 3 \times 3, 256 \\ 3 \times 3, 256 \end{bmatrix} \times 2$ | $\begin{bmatrix} 3 \times 3, 640 \\ 3 \times 3, 640 \end{bmatrix} \times 4$ |
| $(3 \times 3, 512) \times 3$ 
 maxpool, $2 \times 2$ | $\begin{bmatrix} 3 \times 3, 512 \\ 3 \times 3, 512 \end{bmatrix} \times 2$ | |
| fc-4096 
 fc-4096 | avgpool | avgpool |
| fc-10/100 | fc-10/100 | fc-10/100 |

### A.3   IMPLEMENTATION DETAILS

This section provides all the additional implementation details for our experiments.

### A.3.1   MODEL TRAINING

We employ SGD to optimize all the models and the momentum factor is $0.9$. The weight decay factor is set to 5e-4, and the learning rate is decayed by $0.2$ every $50$ epochs.

### A.3.2   ADDITIONAL DETAILS IN SECTION 4.1

We train VGG-16, ResNet-18, Wide-ResNet-28 on CIFAR-10 and CIFAR-100. In the training procedure, the model is trained for 200 epochs, in which the batch size is set to 128, and the learning rate is initialized as $0.1$. There are three training strategies included in this experiment: standard training, non-data-augmentation training, and adversarial training. In the adversarial training, the radius of the adversarial perturbation is set as $10/255$ and $l_\infty$ distance is selected. The basic data augmentation (cropping and flipping) in the standard training and adversarial training is achieved by the following Pytorch code:

```
1 transforms.RandomCrop(32, padding=4)
2 transforms.RandomHorizontalFlip()
```

The experiment is repeated for 10 trials for each (dataset, architecture, training strategy) setting.

### A.3.3   ADDITIONAL DETAILS IN SECTION 4.2

We repeatedly train 10 ResNet-18 on CIFAR-10 and CIFAR-100, respectively, with different random seeds. In the training procedure, the model is trained for 200 epochs, in which the batch size is set to 128, and the learning rate is initialized as $0.1$ and $0.01$, respectively. Basic data augmentation is included during the training process.

### A.3.4   ADDITIONAL DETAILS IN SECTION 4.3

We randomly sample examples from the training set of CIFAR-10 and CIFAR-100 to form five datasets with different sizes of $[2000, 5000, 10000, 20000, 50000]$, respectively. 10 ResNet-18 are trained for each dataset. In the training procedure, the model is trained for 200 epochs, in which the batch size is set to 128, and the learning rate is initialized as $0.1$. Basic data augmentation is included during the training process.

### A.3.5   ADDITIONAL DETAILS IN SECTION 4.4

We randomly change the labels of $20\%$ examples in the training set of CIFAR-10 and CIFAR-100. Then, 10 ResNet-18 are optimize by SGD for 500 epochs on the noise CIFAR-10 and CIFAR-100, respectively. the momentum factor is $0.9$, and the learning rate is $0.001$ and does not decay during the training process.

### A.3.6   ADDITIONAL DETAILS IN SECTION 5.1

We randomly sample examples from the training set of CIFAR-10 and CIFAR-100 to form five datasets with different sizes of $[2000, 5000, 10000, 20000, 50000]$, respectively. For each dataset, we obtain 10 $\eta$-subsets with different $\eta$ of $[0.1, 0.2, 0.3, 0.4, 0.5, 0.6, 0.7, 0.8, 0.9, 1.0]$ via a coreset selection approach named *selection via proxy* (Coleman et al., 2020). The related code can be downloaded from `https://github.com/stanford-futuredata/selection-via-proxy`. The ResNet-18 is repeatedly trained for 10 trials to estimate the complexity of decision boundaries for each $\eta$-subset.

# B    ADDITIONAL DISCUSSION ON DECISION BOUNDARY VARIABILITY

## B.1    ALGORITHM DB VARIABILITY AND THE ENTROPY OF DECISION BOUNDARIES

If Assumption 1 holds for all $\boldsymbol{\theta} \sim \mathbb{Q}$, $1 - AV(f_{\mathbb{Q}}, \mathcal{D})$ can be rewritten as

$$\mathbb{E}_{(\mathbf{x}, y) \sim \mathcal{D}} \mathbb{E}_{\boldsymbol{\theta}, \boldsymbol{\theta}' \sim \mathbb{Q}} \left[ \mathbb{I} \left( T \left( f_{\boldsymbol{\theta}}, \mathbf{x} \right) = T \left( f_{\boldsymbol{\theta}}, \mathbf{x} \right) \right) \right] = \mathbb{E}_{(\mathbf{x}, y) \sim \mathcal{D}} \sum_{i=1}^{k} \mathbb{E}_{\boldsymbol{\theta} \sim \mathbb{Q}}^2 \left[ \mathbb{I} \left( T \left( f_{\boldsymbol{\theta}}, \mathbf{x} \right) = i \right) \right]. \quad (24)$$

The term $\sum_{i=1}^{k} \mathbb{E}_{\boldsymbol{\theta} \sim \mathbb{Q}}^2 \left[ \mathbb{I}(T \left( f_{\boldsymbol{\theta}}, \mathbf{x} \right) = i) \right]$ can be considered to measure the degree of prediction uncertainty for the given voxel $\mathbf{x}$ in the input space $\mathbb{R}^n$. If we leverage $-\log(\cdot)$ on the term $\sum_{i=1}^{k} \mathbb{E}_{\boldsymbol{\theta} \sim \mathbb{Q}}^2 \left[ \mathbb{I}(T \left( f_{\boldsymbol{\theta}}, \mathbf{x} \right) = i) \right]$, then $-\log \sum_{i=1}^{k} \mathbb{E}_{\boldsymbol{\theta} \sim \mathbb{Q}}^2 \left[ \mathbb{I}(T \left( f_{\boldsymbol{\theta}}, \mathbf{x} \right) = i) \right]$ denotes the collision entropy of prediction made by the Gibbs classifier $f_{\mathbb{Q}}$ on $\mathbf{x}$. We can also replace the collision entropy with canonical Shannon entropy $-\sum_{i=1}^{k} \mathbb{E}_{\boldsymbol{\theta} \sim \mathbb{Q}} \left[ \mathbb{I}(T \left( f_{\boldsymbol{\theta}}, \mathbf{x} \right) = i) \right] \log \mathbb{E}_{\boldsymbol{\theta} \sim \mathbb{Q}} \left[ \mathbb{I}(T \left( f_{\boldsymbol{\theta}}, \mathbf{x} \right) = i) \right]$ in the future research. Hence, the algorithm DB variability is closely related to the "entropy of decision boundary", and the uncanny generalization in neural networks might be further uncovered by investigating this low entropy of decision boundary.

## B.2    DATA DB VARIABILITY AND THE COMPLEXITY OF DBS

According to Guan & Loew (2020), a complex decision boundary is considered to own large curvatures and conjectured to indicate inferior generalization. Nevertheless, from the perspective of causality, we argue that the large curvature or non-linearity of DBs *is the result other than the cause* for the classification task, and the primary reason for shaping a complex DB during the training procedure should be the significant non-linearity of the training data. If only the geometric properties of decision boundaries are analysed without investigating the data, the results might be incomplete and even misleading. Another obstacle for describing the complexity of DBs via its geometric properties is the huge dimensional input space, which makes the geometric properties of DBs hard to quantify and estimate. Therefore, defining the complexity of DBs based on its curvature is not rational and impractical.

Here we consider the complexity of DBs from the perspective of the training set. During the training procedure, a small part of training examples, considered as "support vectors", play a more critical role in supervising the formation of decision boundaries and compelling the DB to be gradually more complicated. If the construction of decision boundaries relies on fewer "support vectors", the decision boundary should be simpler. In other words, if these "support vectors" are excluded from the training sample, the DB will be notably dissimilar when the network is retrained on the modified training set. Hence, the complexity of DBs can be also defined with the notion of the data DB variability:

**Definition B.1** (complexity of decision boundaries). Let $f_{\boldsymbol{\theta}}(\mathbf{x}) : \mathbb{R}^n \to \mathbb{R}^k$ be a classifier network parameterized by $\boldsymbol{\theta}$, where $\boldsymbol{\theta} \sim \mathcal{A}(\mathcal{S})$ is returned by leveraging the stochastic learning algorithm $\mathcal{A}$ on the training set $\mathcal{S}$, which is sampled from the data generating distribution $\mathcal{D}$. We term $\mathcal{S}_{\eta} \subset \mathcal{S}$ as a $\eta$-*subset* of $\mathcal{S}$ if $\frac{|\mathcal{S}_{\eta}|}{|\mathcal{S}|} = \eta$. Then, if we fixed $\eta$ and

$$\inf_{\mathcal{S}_{\eta} \subset \mathcal{S}} \mathbb{E}_{(\mathbf{x}, y) \sim \mathcal{D}} \mathbb{E}_{\boldsymbol{\theta} \sim \mathcal{A}(\mathcal{S}), \boldsymbol{\theta}' \sim \mathcal{A}(\mathcal{S}_{\eta})} \left[ \mathbb{I} \left( T(f_{\boldsymbol{\theta}}, \mathbf{x}) \neq T(f_{\boldsymbol{\theta}'}, \mathbf{x}) \right) \right] = \epsilon, \quad (25)$$

the decision boundary of $f_{\mathcal{A}(\mathcal{S})}$ is said to possess a $(\epsilon, \eta)$-*complexity*.

**Implications about the complexity of DB.** With Definition B.1, we could more intuitively understand the relationship between the complexity of DBs and the generalization ability in deep learning: (1) difficult tasks generally have more complex decision boundaries, since their datum are more nonlinear and contain more "support vectors"; (2) in adversarial training, each data point is converted to a "data ball" with the radius of the adversarial perturbation and has more impact on forming the DBs. Hence, adversarial training contributes to a more complex decision boundary by enlarging the "support vector set"; (3) for data augmentation, generated images are also considered to obey the data generation distribution $\mathcal{D}$. Hence, data augmentation decreases the complexity of decision boundaries by greatly expanding the training set $\mathcal{S}$, while $|\mathcal{S}_{\eta}|$ has only a slight growth. Besides, according to our theoretical results in Section 5.2, it can be verified that *neural networks with simpler decision boundaries possess better generalization performance*.

# C  PROOFS

To avoid technicalities, the measurability/integrability issues are ignored throughout this paper. Moreover, Fubini's theorem is assumed to be applicable for any integration *w.r.t.* multiple variables. In other words, the order of integrations is exchangeable. We also use $\mathbb{E}^2[\cdot]$ to denote $[\mathbb{E}[\cdot]]^2$ in our proof for simplicity.

## C.1  PROOF OF THEOREM 1

*Proof.* If Assumption 1 holds for all $\boldsymbol{\theta} \sim \mathbb{Q}$, we have

$$\mathbb{E}_{(\mathbf{x},y)\sim\mathcal{D}}\mathbb{E}_{\boldsymbol{\theta},\boldsymbol{\theta}'\sim\mathbb{Q}}\left[\mathbb{I}\left(y\in T\left(f_{\boldsymbol{\theta}},\mathbf{x}\right)\right)\mathbb{I}\left(y\in T\left(f_{\boldsymbol{\theta}'},\mathbf{x}\right)\right)\mathbb{I}\left(T\left(f_{\boldsymbol{\theta}},\mathbf{x}\right)\neq T\left(f_{\boldsymbol{\theta}'},\mathbf{x}\right)\right)\right]=0 \tag{26}$$

Hence,

$$\begin{aligned} AV(f_{\mathbb{Q}},\mathcal{D}) &=\mathbb{E}_{(\mathbf{x},y)\sim\mathcal{D}}\mathbb{E}_{\boldsymbol{\theta},\boldsymbol{\theta}'\sim\mathbb{Q}}\left[\mathbb{I}\left(T(f_{\boldsymbol{\theta}},\mathbf{x})\neq T(f_{\boldsymbol{\theta}'},\mathbf{x})\right)\right] &(27)\\ &=\mathbb{E}_{(\mathbf{x},y)\sim\mathcal{D}}\mathbb{E}_{\boldsymbol{\theta},\boldsymbol{\theta}'\sim\mathbb{Q}}\left[\mathbb{I}\left(y\in T(f_{\boldsymbol{\theta}},\mathbf{x})\right)\mathbb{I}\left(y\notin T(f_{\boldsymbol{\theta}'},\mathbf{x})\right)\right. &(28)\\ &\quad\left.+\,\mathbb{I}\left(y\notin T(f_{\boldsymbol{\theta}},\mathbf{x})\right)\mathbb{I}\left(T(f_{\boldsymbol{\theta}},\mathbf{x})\neq T(f_{\boldsymbol{\theta}'},\mathbf{x})\right)\right] &(29)\\ &\leq 2\mathcal{R}_{\mathcal{D}}(\mathbb{Q}) &(30) \end{aligned}$$

The proof is completed. $\qquad\square$

## C.2  PROOF OF THEOREM 2

*Proof.* When the classification is binary, *i.e.*, $k = 2$, with Assumption 1, we have

$$\begin{aligned} &\mathbb{E}_{(\mathbf{x},y)\sim\mathcal{D}}\mathbb{E}_{\boldsymbol{\theta},\boldsymbol{\theta}'\sim\mathbb{Q}}\left[\mathbb{I}\left(T\left(f_{\boldsymbol{\theta}},\mathbf{x}\right)=T\left(f_{\boldsymbol{\theta}},\mathbf{x}\right)\right)\right] &(31)\\ =&\mathbb{E}_{(\mathbf{x},y)\sim\mathcal{D}}\mathbb{E}_{\boldsymbol{\theta}\sim\mathbb{Q}}^2\left[\mathbb{I}\left(y\in T\left(f_{\boldsymbol{\theta}},\mathbf{x}\right)\right)\right]+\mathbb{E}_{(\mathbf{x},y)\sim\mathcal{D}}\mathbb{E}_{\boldsymbol{\theta}\sim\mathbb{Q}}^2\left[\mathbb{I}\left(y\notin T\left(f_{\boldsymbol{\theta}},\mathbf{x}\right)\right)\right] &(32)\\ =&\mathbb{E}_{(\mathbf{x},y)\sim\mathcal{D}}\mathbb{E}_{\boldsymbol{\theta}\sim\mathbb{Q}}^2\left[\mathbb{I}\left(y\in T\left(f_{\boldsymbol{\theta}},\mathbf{x}\right)\right)\right]+\mathbb{E}_{(\mathbf{x},y)\sim\mathcal{D}}\left[1-\mathbb{E}_{\boldsymbol{\theta}\sim\mathbb{Q}}\left[\mathbb{I}\left(y\in T\left(f_{\boldsymbol{\theta}},\mathbf{x}\right)\right)\right]\right]^2 &(33)\\ =&2\mathbb{E}_{(\mathbf{x},y)\sim\mathcal{D}}\mathbb{E}_{\boldsymbol{\theta}\sim\mathbb{Q}}^2\left[\mathbb{I}\left(y\in T\left(f_{\boldsymbol{\theta}},\mathbf{x}\right)\right)\right]+2\mathcal{R}_{\mathcal{D}}(\mathbb{Q})-1 &(34) \end{aligned}$$

Plugging in $\mathrm{Var}_{(\mathbf{x},y)\sim\mathcal{D}}\left[\mathbb{E}_{\boldsymbol{\theta}\sim\mathbb{Q}}\left[\mathbb{I}\left(y\in T\left(f_{\boldsymbol{\theta}},\mathbf{x}\right)\right)\right]\right] = \mathbb{E}_{(\mathbf{x},y)\sim\mathcal{D}}\mathbb{E}_{\boldsymbol{\theta}\sim\mathbb{Q}}^2\left[\mathbb{I}\left(y\in T\left(f_{\boldsymbol{\theta}},\mathbf{x}\right)\right)\right] - (1-\mathcal{R}_{\mathcal{D}}(\mathbb{Q}))^2$ yields

$$\begin{aligned} &\mathbb{E}_{(\mathbf{x},y)\sim\mathcal{D}}\mathbb{E}_{\boldsymbol{\theta},\boldsymbol{\theta}'\sim\mathbb{Q}}\left[\mathbb{I}\left(T\left(f_{\boldsymbol{\theta}},\mathbf{x}\right)=T\left(f_{\boldsymbol{\theta}},\mathbf{x}\right)\right)\right] &(35)\\ =&2\mathrm{Var}_{(\mathbf{x},y)\sim\mathcal{D}}\left[\mathbb{E}_{\boldsymbol{\theta}\sim\mathbb{Q}}\left[\mathbb{I}\left(y\in T\left(f_{\boldsymbol{\theta}},\mathbf{x}\right)\right)\right]\right]+\mathcal{R}_{\mathcal{D}}^2(\mathbb{Q})+(1-\mathcal{R}_{\mathcal{D}}(\mathbb{Q}))^2 &(36)\\ \geq&\mathcal{R}_{\mathcal{D}}^2(\mathbb{Q})+(1-\mathcal{R}_{\mathcal{D}}(\mathbb{Q}))^2 &(37) \end{aligned}$$

Plugging in $AV(f_{\mathbb{Q}},\mathcal{D}) = 1 - \mathbb{E}_{(\mathbf{x},y)\sim\mathcal{D}}\mathbb{E}_{\boldsymbol{\theta},\boldsymbol{\theta}'\sim\mathbb{Q}}\left[\mathbb{I}\left(T(f_{\boldsymbol{\theta}},\mathbf{x})=T(f_{\boldsymbol{\theta}'},\mathbf{x})\right)\right]$ and solving the inequality yield the desired inequality and finishes the proof of Theorem 2. $\qquad\square$

### C.3 PROOF OF LEMMA 1

*Proof.* Without loss of generality, we assume that all examples are belong to the first class, *i.e.*, $y = 1$. Hence, for $1 - AV(f_{\mathbb{Q}}, \mathcal{D}) = \mathbb{E}_{(\mathbf{x}, y) \sim \mathcal{D}} \mathbb{E}_{\boldsymbol{\theta}, \boldsymbol{\theta}' \sim \mathbb{Q}} [\mathbb{I} (T (f_{\boldsymbol{\theta}}, \mathbf{x}) = T (f_{\boldsymbol{\theta}'}, \mathbf{x}))]$, we have

$$\mathbb{E}_{(\mathbf{x}, y) \sim \mathcal{D}} \mathbb{E}_{\boldsymbol{\theta}, \boldsymbol{\theta}' \sim \mathbb{Q}} [\mathbb{I} (T (f_{\boldsymbol{\theta}}, \mathbf{x}) = T (f_{\boldsymbol{\theta}'}, \mathbf{x}))] \tag{38}$$

$$= \mathbb{E}_{(\mathbf{x}, y) \sim \mathcal{D}} \sum_{i=1}^{k} \mathbb{E}_{\boldsymbol{\theta} \sim \mathbb{Q}}^{2} [\mathbb{I} (T (f_{\boldsymbol{\theta}}, \mathbf{x}) = i)] \tag{39}$$

$$= \mathbb{E}_{(\mathbf{x}, y) \sim \mathcal{D}} \left[ \mathbb{E}_{\boldsymbol{\theta} \sim \mathbb{Q}}^{2} [\mathbb{I} (T (f_{\boldsymbol{\theta}}, \mathbf{x}) = y)] + \sum_{i=2}^{k} \mathbb{E}_{\boldsymbol{\theta} \sim \mathbb{Q}}^{2} [\mathbb{I} (T (f_{\boldsymbol{\theta}}, \mathbf{x}) = i)] \right] \tag{40}$$

$$= \mathbb{E}_{(\mathbf{x}, y) \sim \mathcal{D}} \left[ \left( 1 - \sum_{i=2}^{k} \mathbb{E}_{\boldsymbol{\theta} \sim \mathbb{Q}} [\mathbb{I} (T (f_{\boldsymbol{\theta}}, \mathbf{x}) = i)] \right)^{2} + \sum_{i=2}^{k} \mathbb{E}_{\boldsymbol{\theta} \sim \mathbb{Q}}^{2} [\mathbb{I} (T (f_{\boldsymbol{\theta}}, \mathbf{x}) = i)] \right] \tag{41}$$

$$\leq 1 + 2 \sum_{i=2}^{k} \mathbb{E}_{(\mathbf{x}, y) \sim \mathcal{D}} \mathbb{E}_{\boldsymbol{\theta} \sim \mathbb{Q}}^{2} [\mathbb{I} (T (f_{\boldsymbol{\theta}}, \mathbf{x}) = i)] - 2 \sum_{i=2}^{k} \mathbb{E}_{(\mathbf{x}, y) \sim \mathcal{D}} \mathbb{E}_{\boldsymbol{\theta} \sim \mathbb{Q}} [\mathbb{I} (T (f_{\boldsymbol{\theta}}, \mathbf{x}) = i)]$$
$$+ 2 \sum_{i > j > 1} \left( \mathbb{E}_{(\mathbf{x}, y) \sim \mathcal{D}} \mathbb{E}_{\boldsymbol{\theta} \sim \mathbb{Q}} [\mathbb{I} (T (f_{\boldsymbol{\theta}}, \mathbf{x}) = i)] \mathbb{E}_{(\mathbf{x}, y) \sim \mathcal{D}} \mathbb{E}_{\boldsymbol{\theta} \sim \mathbb{Q}} [\mathbb{I} (T (f_{\boldsymbol{\theta}}, \mathbf{x}) = j)] \right) \tag{42}$$

$$\leq 1 + 2 \sum_{i=2}^{k} \mathbb{E}_{(\mathbf{x}, y) \sim \mathcal{D}} \mathbb{E}_{\boldsymbol{\theta} \sim \mathbb{Q}}^{2} [\mathbb{I} (T (f_{\boldsymbol{\theta}}, \mathbf{x}) = i)] - 2 \sum_{i=2}^{k} \mathbb{E}_{(\mathbf{x}, y) \sim \mathcal{D}} \mathbb{E}_{\boldsymbol{\theta} \sim \mathbb{Q}} [\mathbb{I} (T (f_{\boldsymbol{\theta}}, \mathbf{x}) = i)]$$
$$+ (k - 2) \sum_{i=2}^{k} \left( \mathbb{E}_{(\mathbf{x}, y) \sim \mathcal{D}} \mathbb{E}_{\boldsymbol{\theta} \sim \mathbb{Q}} [\mathbb{I} (T (f_{\boldsymbol{\theta}}, \mathbf{x}) = i)] \right)^{2} \tag{43}$$

$$\leq 1 + 2 \sum_{i=2}^{k} \mathbb{E}_{(\mathbf{x}, y) \sim \mathcal{D}} \mathbb{E}_{\boldsymbol{\theta} \sim \mathbb{Q}}^{2} [\mathbb{I} (T (f_{\boldsymbol{\theta}}, \mathbf{x}) = i)] - 2 \sum_{i=2}^{k} \mathbb{E}_{(\mathbf{x}, y) \sim \mathcal{D}} \mathbb{E}_{\boldsymbol{\theta} \sim \mathbb{Q}} [\mathbb{I} (T (f_{\boldsymbol{\theta}}, \mathbf{x}) = i)]$$
$$+ (k - 1) \sum_{i=2}^{k} \left( \mathbb{E}_{(\mathbf{x}, y) \sim \mathcal{D}} \mathbb{E}_{\boldsymbol{\theta} \sim \mathbb{Q}} [\mathbb{I} (T (f_{\boldsymbol{\theta}}, \mathbf{x}) = i)] \right)^{2} \tag{44}$$

The penultimate inequality holds because $2 p_i p_j \leq p_i^2 + p_j^2$. Moreover, the variance of $\mathbb{E}_{\boldsymbol{\theta} \sim \mathbb{Q}} [\mathbb{I} (y \notin T (f_{\boldsymbol{\theta}}, \mathbf{x}))]$ is

$$\mathrm{Var}_{(\mathbf{x}, y) \sim \mathcal{D}} [\mathbb{E}_{\boldsymbol{\theta} \sim \mathbb{Q}} [\mathbb{I} (T (f_{\boldsymbol{\theta}}, \mathbf{x}) \neq y)]] \tag{45}$$

$$= \mathbb{E}_{(\mathbf{x}, y) \sim \mathcal{D}} \mathbb{E}_{\boldsymbol{\theta} \sim \mathbb{Q}}^{2} [\mathbb{I} (T (f_{\boldsymbol{\theta}}, \mathbf{x}) \neq y)] - \mathbb{E}_{(\mathbf{x}, y) \sim \mathcal{D}}^{2} \mathbb{E}_{\boldsymbol{\theta} \sim \mathbb{Q}} [\mathbb{I} (T (f_{\boldsymbol{\theta}}, \mathbf{x}) \neq y)], \tag{46}$$

plugging the equality into Eq. 44 yields

$$1 - AV(f_{\mathbb{Q}}, \mathcal{D})$$
$$\leq 1 - (k - 1) \mathrm{Var}_{(\mathbf{x}, y) \sim \mathcal{D}} [\mathbb{E}_{\boldsymbol{\theta} \sim \mathbb{Q}} [\mathbb{I} (T (f_{\boldsymbol{\theta}}, \mathbf{x}) \neq y)]] + (k - 1) \mathbb{E}_{(\mathbf{x}, y) \sim \mathcal{D}} \mathbb{E}_{\boldsymbol{\theta} \sim \mathbb{Q}}^{2} [\mathbb{I} (T (f_{\boldsymbol{\theta}}, \mathbf{x}) \neq y)]$$
$$- (k - 1) \mathbb{E}_{(\mathbf{x}, y) \sim \mathcal{D}}^{2} \mathbb{E}_{\boldsymbol{\theta} \sim \mathbb{Q}} [\mathbb{I} (T (f_{\boldsymbol{\theta}}, \mathbf{x}) \neq y)] + (k - 1) \sum_{i=2}^{k} \mathbb{E}_{(\mathbf{x}, y) \sim \mathcal{D}}^{2} \mathbb{E}_{\boldsymbol{\theta} \sim \mathbb{Q}} [\mathbb{I} (T (f_{\boldsymbol{\theta}}, \mathbf{x}) = i)]$$
$$+ 2 \sum_{i=2}^{k} \mathbb{E}_{(\mathbf{x}, y) \sim \mathcal{D}} \mathbb{E}_{\boldsymbol{\theta} \sim \mathbb{Q}}^{2} [\mathbb{I} (T (f_{\boldsymbol{\theta}}, \mathbf{x}) = i)] - 2 \sum_{i=2}^{k} \mathbb{E}_{(\mathbf{x}, y) \sim \mathcal{D}} \mathbb{E}_{\boldsymbol{\theta} \sim \mathbb{Q}} [\mathbb{I} (T (f_{\boldsymbol{\theta}}, \mathbf{x}) = i)] \tag{47}$$

Then, because $\mathbb{E}_{\boldsymbol{\theta} \sim \mathbb{Q}} [\mathbb{I} (T (f_{\boldsymbol{\theta}}, \mathbf{x}) \neq y)] = \sum_{i=2}^{k} \mathbb{E}_{\boldsymbol{\theta} \sim \mathbb{Q}} [\mathbb{I} (T (f_{\boldsymbol{\theta}}, \mathbf{x}) = i)]$, we have

$$\sum_{i=2}^{k} \mathbb{E}_{(\mathbf{x}, y) \sim \mathcal{D}}^{2} \mathbb{E}_{\boldsymbol{\theta} \sim \mathbb{Q}} [\mathbb{I} (T (f_{\boldsymbol{\theta}}, \mathbf{x}) = i)] \leq \mathbb{E}_{(\mathbf{x}, y) \sim \mathcal{D}}^{2} \mathbb{E}_{\boldsymbol{\theta} \sim \mathbb{Q}} [\mathbb{I} (T (f_{\boldsymbol{\theta}}, \mathbf{x}) \neq y)] \tag{48}$$

and

$$\sum_{i=2}^{k} \mathbb{E}_{(\mathbf{x}, y) \sim \mathcal{D}} \mathbb{E}_{\boldsymbol{\theta} \sim \mathbb{Q}}^{2} [\mathbb{I} (T (f_{\boldsymbol{\theta}}, \mathbf{x}) = i)] \leq \mathbb{E}_{(\mathbf{x}, y) \sim \mathcal{D}} \mathbb{E}_{\boldsymbol{\theta} \sim \mathbb{Q}}^{2} [\mathbb{I} (T (f_{\boldsymbol{\theta}}, \mathbf{x}) \neq y)] . \tag{49}$$

Therefore, Eq. 47 can be scaled to

$$(k-1)\text{Var}_{(\mathbf{x},y)\sim\mathcal{D}}\left[\mathbb{E}_{\boldsymbol{\theta}\sim\mathbb{Q}}\left[\mathbb{I}\left(T\left(f_{\boldsymbol{\theta}},\mathbf{x}\right)\neq y\right)\right]\right]$$

$$\leq AV(f_{\mathbb{Q}},\mathcal{D})+(k+1)\mathbb{E}_{(\mathbf{x},y)\sim\mathcal{D}}\mathbb{E}_{\boldsymbol{\theta}\sim\mathbb{Q}}^2\left[\mathbb{I}\left(T\left(f_{\boldsymbol{\theta}},\mathbf{x}\right)\neq y\right)\right]-2\sum_{i=2}^{k}\mathbb{E}_{(\mathbf{x},y)\sim\mathcal{D}}\mathbb{E}_{\boldsymbol{\theta}\sim\mathbb{Q}}\left[\mathbb{I}\left(T\left(f_{\boldsymbol{\theta}},\mathbf{x}\right)=i\right)\right]$$

$$(50)$$

Then, with Assumption 2 that $\mathbb{E}_{(\mathbf{x},y)\sim\mathcal{D}}\mathbb{E}_{\boldsymbol{\theta}\sim\mathbb{Q}}^2\left[\mathbb{I}\left(y\notin T\left(f_{\boldsymbol{\theta}},\mathbf{x}\right)\right)\right]\leq\frac{2}{k+1}\mathcal{R}_{\mathcal{D}}(\mathbb{Q})$, we get the following inequality *w.r.t.* the variance:

$$\text{Var}_{(\mathbf{x},y)\sim\mathcal{D}}\left[\mathbb{E}_{\boldsymbol{\theta}\sim\mathbb{Q}}\left[\mathbb{I}\left(T\left(f_{\boldsymbol{\theta}},\mathbf{x}\right)\neq y\right)\right]\right]\leq\frac{AV(f_{\mathbb{Q}},\mathcal{D})}{k-1}. \tag{51}$$

The proof is completed. □

## C.4 PROOF OF THEOREM 3

*Proof.* For the random variable $Z_i$, recall the *one-sided* Chebyshev's inequality that

$$\Pr\left[\mathbb{E}[Z_1]-\frac{1}{m}\sum_{i=1}^{m}Z_i>a\right]\leq\frac{\text{Var}[Z_1]}{2ma^2}. \tag{52}$$

Let $Z_i=\mathbb{E}_{\boldsymbol{\theta}\sim\mathbb{Q}}\left[\mathbb{I}\left(y_i\notin T\left(f_{\boldsymbol{\theta}},\mathbf{x}_i\right)\right)\right]$, so we have

$$\Pr\left[\mathcal{R}_{\mathcal{D}}(\mathbb{Q})-\mathcal{R}_S(\mathbb{Q})>a\right]\leq\frac{\text{Var}_{(\mathbf{x},y)\sim\mathcal{D}}\left[\mathbb{E}_{\boldsymbol{\theta}\sim\mathbb{Q}}\left[\mathbb{I}\left(y\notin T\left(f_{\boldsymbol{\theta}},\mathbf{x}\right)\right)\right]\right]}{2ma^2}. \tag{53}$$

Plugging Lemma 1 into the above inequality yields

$$\Pr\left[\mathcal{R}_{\mathcal{D}}(\mathbb{Q})-\mathcal{R}_S(\mathbb{Q})>a\right]\leq\frac{AV(f_{\mathbb{Q}},\mathcal{D})}{2m(k-1)a^2}. \tag{54}$$

Plugging in $\delta=\frac{AV(f_{\mathbb{Q}},\mathcal{D})}{2m(k-1)a^2}$ yields the desired inequality and completes the proof. □

## C.5 PROOF OF LEMMA 2

*Proof.* From the definition of the complexity of DB, there exists a $\eta$-subset $\mathcal{S}_\eta$ s.t.

$$\mathbb{E}_{(\mathbf{x},y)\sim\mathcal{D}}\mathbb{E}_{\boldsymbol{\theta}\sim\mathcal{A}(\mathcal{S}),\boldsymbol{\theta}'\sim\mathcal{A}(\mathcal{S}_\eta)}\left[\mathbb{I}\left(T(f_{\boldsymbol{\theta}},\mathbf{x})\neq T(f_{\boldsymbol{\theta}'},\mathbf{x})\right)\right]=\epsilon. \tag{55}$$

Recall the LHS $=|\mathcal{R}_{\mathcal{D}}(\mathcal{A}(\mathcal{S}))-\mathcal{R}_{\mathcal{D}}(\mathcal{A}(\mathcal{S}_\eta))|$, then

$$\text{LHS}=\left|\mathbb{E}_{(\mathbf{x},y)\sim\mathcal{D}}\mathbb{E}_{\boldsymbol{\theta}\sim\mathcal{A}(\mathcal{S})}\left[\mathbb{I}\left(y\notin T(f_{\boldsymbol{\theta}},\mathbf{x})\right)\right]-\mathbb{E}_{(\mathbf{x},y)\sim\mathcal{D}}\mathbb{E}_{\boldsymbol{\theta}\sim\mathcal{A}(\mathcal{S}_\eta)}\left[\mathbb{I}\left(y\notin T(f_{\boldsymbol{\theta}},\mathbf{x})\right)\right]\right| \tag{56}$$

$$=\left|\mathbb{E}_{(\mathbf{x},y)\sim\mathcal{D}}\mathbb{E}_{\boldsymbol{\theta}\sim\mathcal{A}(\mathcal{S}),\boldsymbol{\theta}'\sim\mathcal{A}(\mathcal{S}_\eta)}\left[\mathbb{I}\left(y\notin T(f_{\boldsymbol{\theta}},\mathbf{x})\right)-\mathbb{I}\left(y\notin T(f_{\boldsymbol{\theta}'},\mathbf{x})\right)\right]\right| \tag{57}$$

$$\leq\mathbb{E}_{(\mathbf{x},y)\sim\mathcal{D}}\mathbb{E}_{\boldsymbol{\theta}\sim\mathcal{A}(\mathcal{S}),\boldsymbol{\theta}'\sim\mathcal{A}(\mathcal{S}_\eta)}\left[\left|\mathbb{I}\left(y\notin T(f_{\boldsymbol{\theta}},\mathbf{x})\right)-\mathbb{I}\left(y\notin T(f_{\boldsymbol{\theta}'},\mathbf{x})\right)\right|\right] \tag{58}$$

$$\leq\mathbb{E}_{(\mathbf{x},y)\sim\mathcal{D}}\mathbb{E}_{\boldsymbol{\theta}\sim\mathcal{A}(\mathcal{S}),\boldsymbol{\theta}'\sim\mathcal{A}(\mathcal{S}_\eta)}\left[\mathbb{I}\left(T(f_{\boldsymbol{\theta}},\mathbf{x})\neq T(f_{\boldsymbol{\theta}'},\mathbf{x})\right)\right]=\epsilon \tag{59}$$

The proof is completed. □

## C.6 PROOF OF LEMMA 3

*Proof.* Let

$$\mu=\mathbb{E}_{(\mathbf{x},y)\sim\mathcal{D}}\mathbb{E}_{\boldsymbol{\theta}\sim\mathcal{A}(\mathcal{S}),\boldsymbol{\theta}'\sim\mathcal{A}(\mathcal{S}_\eta)}\left[\mathbb{I}\left(T(f_{\boldsymbol{\theta}},\mathbf{x})\neq T(f_{\boldsymbol{\theta}'},\mathbf{x})\right)\right] \tag{60}$$

$$\hat{\mu}=\frac{1}{m}\sum_{i=1}^{m}\mathbb{E}_{\boldsymbol{\theta}\sim\mathcal{A}(\mathcal{S}),\boldsymbol{\theta}'\sim\mathcal{A}(\mathcal{S}_\eta)}\left[\mathbb{I}\left(T(f_{\boldsymbol{\theta}},\mathbf{x}_i)\neq T(f_{\boldsymbol{\theta}'},\mathbf{x}_i)\right)\right], \tag{61}$$

By applying Hoeffding's Inequality, we have

$$\Pr\left[\hat{\mu}-\mu\geq t\right]\leq\exp(-2mt^2). \tag{62}$$

Plug in $\delta=\exp(-2mt^2)$ into Eq. 62, thus, with the probability of at least $1-\delta$, we have

$$\hat{\mu}\leq\sqrt{\frac{1}{2m}\log\frac{1}{\delta}}+\mu\leq\sqrt{\frac{1}{2m}\log\frac{1}{\delta}}+\epsilon. \tag{63}$$

Following the proof of Lemma 2, we can derive the inequality of $|\mathcal{R}_S(\mathcal{A}(\mathcal{S}_\eta))-\mathcal{R}_S(\mathcal{A}(\mathcal{S}))|\leq\hat{\mu}$. Plugging this into Eq. 63 yields the desired inequality and concludes the proof of Lemma 3. □

### C.7 Proof of Theorem 4

We first introduce Lemma 4 and Lemma 5 as below.

**Lemma 4** (Lemma 30.1 in (Shalev-Shwartz & Ben-David, 2014)). Assume $T$ and $V$ are two datasets independently sampled from the data generating distribution $\mathcal{D}$, then, with the probability of at least $1 - \delta$, we have

$$\mathcal{R}_\mathcal{D}(\mathcal{A}(T)) \leq \mathcal{R}_V(\mathcal{A}(T)) + \sqrt{\frac{2\mathcal{R}_V(\mathcal{A}(T))\log(1/\delta)}{|V|}} + \frac{4\log(1/\delta)}{|V|}. \tag{64}$$

**Lemma 5** (Theorem 30.2 in (Shalev-Shwartz & Ben-David, 2014)). Let $\mathcal{S}_\eta$ be a $\eta$-subnet of the dataset $\mathcal{S}$, which is sampled from the data generation distribution $\mathcal{D}$ and the sample size $|\mathcal{S}| = m$. Let $\mathcal{S}\backslash\mathcal{S}_\eta = \mathcal{S} - \mathcal{S}_\eta$ and assume $\eta \leq 0.5$. Then, with the probability of at least $1 - \delta$ over a sample of size $m$, we have

$$\mathcal{R}_\mathcal{D}(\mathcal{A}(\mathcal{S}_\eta)) \leq \mathcal{R}_{\mathcal{S}\backslash\mathcal{S}_\eta}(\mathcal{A}(\mathcal{S}_\eta)) + \sqrt{4\mathcal{R}_{\mathcal{S}\backslash\mathcal{S}_\eta}(\mathcal{A}(\mathcal{S}_\eta))\Delta} + 8\Delta, \tag{65}$$

where

$$\Delta = \eta\log\frac{e}{\eta} + \frac{1}{m}\log\frac{1}{\delta'}. \tag{66}$$

*Proof of Lemma 5.*

$$\Pr\left[\exists\mathcal{S}_\eta \subseteq \mathcal{S} \text{ s.t. } \mathcal{R}_\mathcal{D}(\mathcal{A}(\mathcal{S}_\eta)) \leq \mathcal{R}_{\mathcal{S}\backslash\mathcal{S}_\eta}(\mathcal{A}(\mathcal{S}_\eta)) + \sqrt{\frac{2\mathcal{R}_{\mathcal{S}\backslash\mathcal{S}_\eta}(\mathcal{A}(\mathcal{S}_\eta))\log(1/\delta)}{|\mathcal{S}\backslash\mathcal{S}_\eta|}} + \frac{4\log(1/\delta)}{|\mathcal{S}\backslash\mathcal{S}_\eta|}\right] \tag{67}$$

$$\leq \sum_{\mathcal{S}_\eta \subseteq \mathcal{S}} \Pr\left[\mathcal{R}_\mathcal{D}(\mathcal{A}(\mathcal{S}_\eta)) \leq \mathcal{R}_{\mathcal{S}\backslash\mathcal{S}_\eta}(\mathcal{A}(\mathcal{S}_\eta)) + \sqrt{\frac{2\mathcal{R}_{\mathcal{S}\backslash\mathcal{S}_\eta}(\mathcal{A}(\mathcal{S}_\eta))\log(1/\delta)}{|\mathcal{S}\backslash\mathcal{S}_\eta|}} + \frac{4\log(1/\delta)}{|\mathcal{S}\backslash\mathcal{S}_\eta|}\right] \tag{68}$$

$$= \binom{m}{\eta m}\delta \leq \left(\frac{e}{\eta}\right)^{\eta m}\delta \tag{69}$$

Plug in $\delta' = \left(\frac{e}{\eta}\right)^{\eta m}\delta$, and use the assumption $\eta \leq \frac{1}{2}$, which implies $|\mathcal{S}\backslash\mathcal{S}_\eta| \geq \frac{m}{2}$, then, with the probability of at least $1 - \delta'$, we have that

$$\mathcal{R}_\mathcal{D}(\mathcal{A}(\mathcal{S}_\eta)) \leq \mathcal{R}_{\mathcal{S}\backslash\mathcal{S}_\eta}(\mathcal{A}(\mathcal{S}_\eta)) + \sqrt{4\mathcal{R}_{\mathcal{S}\backslash\mathcal{S}_\eta}(\mathcal{A}(\mathcal{S}_\eta))\left(\eta\log\frac{e}{\eta} + \frac{1}{m}\log\frac{1}{\delta'}\right)} + 8\left(\eta\log\frac{e}{\eta} + \frac{1}{m}\log\frac{1}{\delta'}\right), \tag{70}$$

which concludes the proof. □

With the above lemmas, we can derive the generalization bound based on the complexity of decision boundary.

*Proof of Theorem 4.* With the assumption of zero-training error that $\mathcal{R}_\mathcal{S}(\mathcal{A}(\mathcal{S})) = \mathcal{R}_{\mathcal{S}_\eta}(\mathcal{A}(\mathcal{S}_\eta)) = 0$, we can derive the follow inequality:

$$\mathcal{R}_{\mathcal{S}\backslash\mathcal{S}_\eta}(\mathcal{A}(\mathcal{S}_\eta)) = \frac{1}{1-\eta}\left(\mathcal{R}_\mathcal{S}(\mathcal{A}(\mathcal{S}_\eta)) - \mathcal{R}_{\mathcal{S}_\eta}(\mathcal{A}(\mathcal{S}_\eta))\right) = \frac{1}{1-\eta}\mathcal{R}_\mathcal{S}(\mathcal{A}(\mathcal{S}_\eta)) \tag{71}$$

Then, by plugging the above equation into to Lemma 3, with the probability of at least $1 - \delta$, we have

$$\mathcal{R}_{\mathcal{S}\backslash\mathcal{S}_\eta}(\mathcal{A}(\mathcal{S}_\eta)) \leq \frac{1}{1-\eta}\left(\sqrt{\frac{1}{2m}\log\frac{1}{\delta}} + \epsilon\right) \tag{72}$$

Through combining this with Lemma 5, with the probability of at least $1 - 2\delta$, we have

$$\mathcal{R}_\mathcal{D}(\mathcal{A}(\mathcal{S}_\eta)) \leq \Omega + \sqrt{4\Omega\Delta} + 8\Delta, \tag{73}$$

where

$$\Omega = \frac{1}{1-\eta} \left( \sqrt{\frac{1}{2m} \log \frac{1}{\delta}} + \epsilon \right), \tag{74}$$

$$\Delta = \eta \log \frac{e}{\eta} + \frac{1}{m} \log \frac{1}{\delta}. \tag{75}$$

Plugging the equality of $\mathcal{R}_\mathcal{D}\left(\mathcal{A}\left(\mathcal{S}\right)\right) \le \mathcal{R}_\mathcal{D}\left(\mathcal{A}\left(\mathcal{S}_\eta\right)\right) + \epsilon$ in Lemma 2 into Eq. 73, with the probability of at least $1 - 2\delta$, we have

$$\mathcal{R}_\mathcal{D}(\mathcal{A}(\mathcal{S})) \le \Omega + \sqrt{4\Omega\Delta} + 8\Delta + \epsilon. \tag{76}$$

Plugging in $\delta' = 2\delta$ yields the desired inequality of Eq. 20.

When $m$ is sufficient large, $\sqrt{4\Omega\Delta}$ can be dropped due to $\sqrt{4\Omega\Delta} \le \Omega + \Delta$. Considering $\eta \le 0.5$, $\Omega \le 2 \left( \sqrt{\frac{1}{2m} \log \frac{1}{\delta}} + \epsilon \right) = \mathcal{O}(\frac{1}{\sqrt{m}} + \epsilon)$. The term $\frac{1}{m} \log \frac{1}{\delta}$ in $\Delta$ can be dropped because it has a faster convergence speed compared to $\sqrt{\frac{1}{2m} \log \frac{1}{\delta}}$ in $\Omega$. Because $\log \frac{1}{\delta}$ is considered as a constant, we have

$$\mathcal{R}_\mathcal{D}(\mathcal{A}(\mathcal{S})) \le \mathcal{O}(\frac{1}{\sqrt{m}} + \epsilon + \eta \log \frac{1}{\eta}). \tag{77}$$

The proof of Theorem 4 is finished. $\qquad\square$

## D  ADDITIONAL EXPERIMENTAL RESULTS

This appendix collect experimental results omitted from the main text due to the space limitation.

### D.1  ADDITIONAL RESULTS FOR LEARNING RATE

Accroding to Li et al. (2019) that learning rate is a major factor of affecting generalization of networks, we investigate the relationship between algorithm DB variability and learning rate by training 20 ResNet-18 with different constant learning rates (no decay) of 0.001 and 0.0001 on CIFAR-10 until the training procedure converges. Then, the average test error and algorithm DB variability are calculated at each epoch, as shown in Figure 5. From the plots, we observe that: (1) the larger learning rate 0.001 contributes to better generalization performance; and (2) algorithm DB variability with the larger learning rate 0.001 is smaller during the training process, compared to the learning rate 0.0001. Therefore, there is still a negative correlation between algorithm DB variability and generalization by varying the learning rate, and thus Hypothesis 1 is well supported.

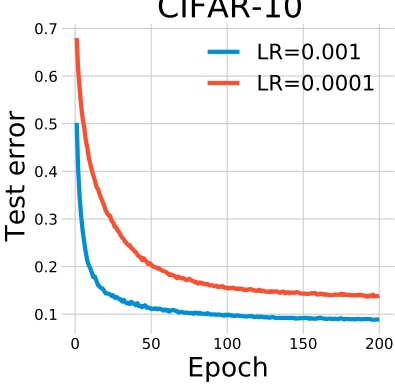 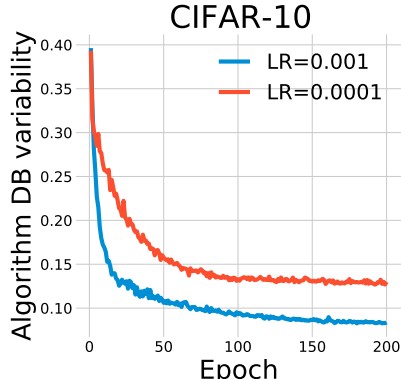

(a) Test error vs. training time  (b) Algorithm DB variability vs. training time

Figure 5: (a) Plots of test error as a function of training time (LR is learning rate). (b) Plots of algorithm DB variability as a function of training time (LR is learning rate). Each curve is calculated and then averaged on 10 trials.

