# OpenReview forum: "Decision boundary variability and generalization in neural networks"
_ICLR.cc/2022/Conference — ICLR 2022 Submitted_

### Official Review · Reviewer_kiZ8 · 2021-11-01

**Correctness:** 4
**Technical Novelty And Significance:** 3
**Empirical Novelty And Significance:** 3
**Recommendation:** 6
**Confidence:** 4

**Main Review:**

My main concern about this paper is how well the findings hold in a more realistic setting, given the strong assumptions made in the limited experiments and several confusions in the theoretical part.

1. Although the authors see a certain level of negative correlation between the decision boundary (DB) variability and generalization, the variability may not be a reliable characterization of generalization unless the following confounding factors are well addressed:

   * Training steps: models trained with different numbers of steps may have different variabilities but the same generalizability. for example, the claim that algorithm DB variability is zero on the linearly separable data is not true unless SGD takes infinite steps and other assumptions hold [1]. So training with different numbers of training steps, the variability would change. While after a certain amount of steps, which is often observed empirically, the generalization gap stops changing. In this case, the variability does not well reflect the generalization - more experiments are needed to understand this.

    * Label noise: continue with the training steps, if there exists some level of label noises and consequently epoch-wise double descent [2], the connection between training steps and generalization is even more complicated. Does the DB variability still characterize the generalizability?

    * Factors that may increase the DB variability while improving the generalization: there are cases where instabilities of the training process leading to better generalization, for example, dropout and large learning rate [3]. Do such instabilities also lead to larger DB variability? If so, why is better generalization achieved?

    * The current observations on the negative correlation between DB variability and generalizability can probably be both explained by over-parameterization. And in the other cases like those listed above, it is unclear whether the correlation still holds.

2. The quantification of the DB variability depends on the data generating distribution, which is however vaguely defined in the paper. For the two-moon dataset, the data generating distribution seems to be the $\mathcal{R}^2$, and the authors empirically selected the rectangular grid in their experiments. But on the Cifar-10 dataset, the author picked the images generated by BigGAN as the data generating distribution, where the decision boundaries off the data manifold are excluded.

3. The authors propose two factors, simplicity effect, and constraint effect, to account for the algorithm DB variability without rigorous definitions which have led to confusion. For example, when discussing the adversarial training under section 4.1, the paper says "which (adversarial training) substantially decreases the inter-class distances and makes the training set more complex", the authors seem to confuse the two factors into the same phenomenon. It is not convincing to me that significant the difference between these two factors is.

4.  The motivation of the data DB variability is not clear to me. The authors motivate the data DB variability as "the algorithm DB
variability hardly shows the decision boundary variability caused by changes in training data". Why is that important in characterizing the generalizability? And the paper lacks the comparison between the bounds built upon the two variability metrics. It is not clear to me when one is more useful / tighter than the other and should be used.

Minor:

1. "Remark 1. (1) If a input..." -> "Remark 1. (1) If an input..."

[1] Soudry, Daniel, et al. "The implicit bias of gradient descent on separable data." The Journal of Machine Learning Research 19.1 (2018): 2822-2878.

[2] Nakkiran, Preetum, et al. "Deep Double Descent: Where Bigger Models and More Data Hurt." International Conference on Learning Representations. 2019.

[3] Li, Yuanzhi, Colin Wei, and Tengyu Ma. "Towards Explaining the Regularization Effect of Initial Large Learning Rate in Training Neural Networks." Advances in Neural Information Processing Systems 32 (2019).

**Summary Of The Paper:**

This paper proposes to characterize the generalizability with the variability of decision boundary instead of the margin. They show empirically that generalizability is negatively correlated with the variability and theoretically bounded by it.

**Summary Of The Review:**

For the reasons listed above, I think substantial experiments are required to better understand the phenomenon and confirm the empirical findings, rigorous definitions and clarifications are needed in the theoretical sections, and thereby my current rating.

---

> ### Author Response · Authors · 2021-11-19
> **To Reviewer kiZ8 (3/3)**
>
> **Q3:** *The authors propose two factors, simplicity effect, and constraint effect, to account for the algorithm DB variability without rigorous definitions which have led to confusion. For example, when discussing the adversarial training under section 4.1, the paper says "which (adversarial training) substantially decreases the inter-class distances and makes the training set more complex", the authors seem to confuse the two factors into the same phenomenon. It is not convincing to me that significant the difference between these two factors is.*
>
> **A3:** Thanks for your suggestion. We have carefully revised our presentation to make it clear. The terms, simplicity effect and constraint effect, have been removed.
>
> **Q4:** *The motivation of the data DB variability is not clear to me. The authors motivate the data DB variability as "the algorithm DB variability hardly shows the decision boundary variability caused by changes in training data". Why is that important in characterizing the generalizability?*
>
> **A4:** We find that decision boundary variability can characterizes generalizability. To further understand this characterization, we decompose decision boundary variability into algorithm DB variability and data DB variability, in order to decomposes the impact from algorithm and data.
>
> Intuitively, the training process in deep learning relies on stochastic gradients, which also bring significant randomness into prediction of the learning model, or equivalently, the decision boundary in the classification task. Similarly, as a finite sample drawn from the data generating distribution, the training data bring considerable fluctuation into the decision boundary. These two phenomena are characterized by algorithm DB variability and data DB variability, respectively.
>
> As shown in many existing works, including [1], the randomness of prediction may characterize the generalizability of neural networks. Therefore, algorithm DB variability and data DB variability would be good indicators of the generalizability.
>
>   We have added a more detailed discussion in Appendix B.2.
>
> **Q5:** *The paper lacks the comparison between the bounds built upon the two variability metrics. It is not clear to me when one is more useful / tighter than the other and should be used.*
>
> **A5:** Thanks.  We have added detailed discussion and comparison between the two upper bounds (Theorem 3, Theorem 4).
>
>   The bounds are obtained from algorithm decision boundary variability and data decision boundary variability, and are of order $\mathcal{O}\left(\sqrt{AV(f_{\mathbb{Q}}, \mathcal{D})/m(k-1)}\right)$ and $\mathcal{O}\left(\frac{1}{\sqrt{m}}+\epsilon+\eta\log\frac{1}{\eta}\right)$, respectively, where $m$ is the sample size, $k$ is the number of classes, $AV(f_{\mathbb{Q}}, \mathcal{D})$ is the algorithm DB variability, and $(\epsilon,\eta)$ are parameters of data DB variability.
>
> The bounds are obtained from algorithm DB variability and data DB variability, respectively. They are of orders $\mathcal{O}\left(\sqrt{AV(f_{\mathbb{Q}}, \mathcal{D})/m(k-1)}\right)$ and $\mathcal{O}\left(\frac{1}{\sqrt{m}}+\epsilon+\eta\log\frac{1}{\eta}\right)$, where $m$ is the sample size, $k$ is the number of classes, $AV(f_{\mathbb{Q}}, \mathcal{D})$ is the algorithm DB variability, and $(\epsilon,\eta)$ are parameters of data DB variability. Detailed comparisons between them are given below:
>
> 1.	Theorem 3 and Theorem 4 characterize the impact from algorithm and data on the generalizability, respectively;
> 2.	Compared with Theorem 4, Theorem 3 relies on additional Assumptions 2 and 3 (which have been relaxed to a new weaker Assumption 2; please kindly refer to A3 to Reviewer pU8o, A1.1 to Reviewer 3dWe, and Section 4.5.2 in the revised manuscript for more details);
> 3.	Theorem 3 can better characterize the impact of the class number $k$ on the generalizability;
> 4.	Theorem 3 can be better empirically calculated: algorithm DB variability $AV(f_\mathbb{Q},\mathcal{D})$ is easier to calculate than data DB variability $(\epsilon,\eta)$ in practice.
>
> **Q6:** *"Remark 1. (1) If a input..." -> "Remark 1. (1) If an input..."*
>
>   **A6:** Thanks and addressed.
>
> [1] T. Poggio, Q. Liao, A. Banburski. “Complexity control by gradient descent in deep networks.” *Nature communications*. 11(1):1-5. 2020.

---

> ### Author Response · Authors · 2021-11-19
> **To Reviewer kiZ8 (2/3)**
>
>
> **Q1.3:** *Factors that may increase the DB variability while improving the generalization: there are cases where instabilities of the training process leading to better generalization, for example, dropout and large learning rate. Do such instabilities also lead to larger DB variability? If so, why is better generalization achieved?*
>
> **A1.3:**  Thanks. We conducted experiments following your suggestion, which fully support our arguments.
>
> We train ResNet-18 on CIFAR-10 and CIFAR-100 with different initial learning rates of $0.1$ and $0.01$.  Based on the learned models, we calculate the algorithm DB variability and test error at each epoch of the training process as below; please kindly refer to A1.1 for the results. The experimental results show that the initial learning rates have no significant influence on neither algorithm DB variability nor generalization performance. Besides, a clear positive correlation is observed between algorithm DB variability and test error, which fully supports the claims of our paper. Please kindly refer to Section 4.2 (Figure 2) in the revised manuscript for more details.
>
> Besides, we sincerely note that dropout is not so effective in popular architectures, such as ResNet. Thus, we conduct additional experiments to explore the relationship between algorithm DB variability and training sample size, which is a more important factor that affects the generalization ability of networks. The results fully support our claims.
>
> We train ResNet-18 with different sample size of $[2,000,\ 5,000,\ 10,000,\ 20,000,\ 50,000]$ on CIFAR-10 and CIFAR-100. Based on these models, we calculate algorithm DB variability and test error as below:
>
>   - **CIFAR-10**
>
>   | Sample size | 2,000 | 5,000 | 10,000 | 20,000 | 50,000 |
>   | :---------: | :--: | :--: | :---: | :---: | :---: |
>   | Test error  | 0.44 | 0.27 | 0.15  | 0.09  | 0.05  |
>   | Algorithm DB variability| 0.33 | 0.19 | 0.11  | 0.07  | 0.05  |
>
>   - **CIFAR-100**
>
>   | Sample size | 2,000 | 5,000 | 10,000 | 20,000 | 50,000 |
>   | :---------: | :--: | :--: | :---: | :---: | :---: |
>   | Test error  | 0.82 | 0.67 | 0.48  | 0.33  | 0.22  |
>   | Algorithm DB variability| 0.57 | 0.48 | 0.36  | 0.26  | 0.21  |
>
> From the tables, a clear positive correlation is observed between algorithm DB variability and test error, which fully supports the claims of our paper. Please kindly refer to Section 4.3 (Figure 3(a)) in the revised manuscript for more details.
>
>
> **Q1.4:** *The current observations on the negative correlation between DB variability and generalizability can probably be both explained by over-parameterization. And in the other cases like those listed above, it is unclear whether the correlation still holds.*
>
> **A1.4:** Thanks. According to the above empirical results, the close correlation between DB variability and generalization holds in all scenarios, which fully support the claims of our paper.
>
> **Q2.1**: *The quantification of the DB variability depends on the data generating distribution, which is however vaguely defined in the paper. For the two-moon dataset, the data generating distribution seems to be the R2, and the authors empirically selected the rectangular grid in their experiments. On the Cifar-10 dataset, the author picked the images generated by BigGAN as the data generating distribution, where the decision boundaries off the data manifold are excluded.*
>
> **A2.1:** Thanks for this question! The data generating distribution is defined to be the latent distribution of data. To avoid any ambiguity, we have removed the parts for two-moon dataset according to your suggestions. In the rebuttal session, we have added additional experiments as shown above.

---

> > ### Comment · Reviewer_kiZ8 · 2021-11-20
> > **Thank you for your response**
> >
> > The large initial learning rate experiment is bit different from what I asked: as shown in [1], a large initial learning rate with scheduled decay generalizes better than a small learning rate without decay. There might not be enough time at the rebuttal period but I would recommend the authors to add the experiment with learning rate = 0.001 and no learning schedule in their future version. This should give a worse generalization, and probably a larger variability.
> >
> > Minor: What are the difference of the dots in Figure 2 (b)? Are they from different epochs? You may want to make this clear in the caption or Section 4.2.
> >
> > [1] Li, Yuanzhi, Colin Wei, and Tengyu Ma. "Towards Explaining the Regularization Effect of Initial Large Learning Rate in Training Neural Networks." Advances in Neural Information Processing Systems 32 (2019).

---

> > > ### Author Response · Authors · 2021-11-21
> > > **Thanks for your suggestion!**
> > >
> > > Thanks for your suggestion! We will add the experiments according to your suggestion in the future version.
> > >
> > > To minor: Thanks. The dots in Figure 2 (b) are collected from different epochs. We will clearly state this in the revised version.

---

> > > ### Author Response · Authors · 2021-11-22
> > > **Update: additional experiment on learning rate ready**
> > >
> > > We train $20$ ResNet-18 with different constant learning rates (without decay) of $0.001$ and $0.0001$ on CIFAR-10. Based on the learned models, we calculate the algorithm DB variability and test error at each epoch of the training process. Then, we draw curves of test error and algorithm DB variability curve with respect to the learning rate. These curves show that smaller learning rate corresponds to worse generalization performance and larger algorithm DB variability, which are fully aligned with your conjecture (and our claims).
> > >
> > > Please kindly refer the Appendix D.1 (Figure 5) in the revised manuscript for more details.

---

> > > > ### Comment · Reviewer_kiZ8 · 2021-11-22
> > > > **Reply**
> > > >
> > > > Thank you for your update on the additional experiments. I will keep my score.

---

> ### Author Response · Authors · 2021-11-19
> **To Reviewer kiZ8 (1/3)**
>
> Thank you for your thorough review and constructive feedback. All your concerns have been carefully responded. The manuscript is carefully revised accordingly. We sincerely hope our responses fully address your questions.
>
> **Q1.1:** *Training steps: models trained with different numbers of steps may have different variabilities but the same generalizability. For example, the claim that algorithm DB variability is zero on the linearly separable data is not true unless SGD takes infinite steps and other assumptions hold [1]. So training with different numbers of training steps, the variability would change. While after a certain amount of steps, which is often observed empirically, the generalization gap stops changing. In this case, the variability does not well reflect the generalization - more experiments are needed to understand this.*
>
> **A1.1:** Thanks. We conducted experiments following your suggestion, which fully support our arguments.
>
> We train ResNet-18 on CIFAR-10 with initial learning rates of $0.1$ and $0.01$.  Based on the learned models, we calculate the algorithm DB variability and test error at each epoch of the training process as below,
>
>   - **Initial learning rate=0.1**
>
>     **(CIFAR-10)**
>
>     |      Training epoch      |  1   |  3   |  10  |  30  | 100  | 200  |
>     | :----------------------: | :--: | :--: | :--: | :--: | :--: | :--: |
>     |        Test error        | 0.58 | 0.41 | 0.22 | 0.19 | 0.09 | 0.05 |
>     | Algorithm DB variability | 0.50 | 0.39 | 0.23 | 0.22 | 0.12 | 0.05 |
>
>     **(CIFAR-100)**
>
>     |      Training epoch      |  1   |  3   |  10  |  30  | 100  | 200  |
>     | :----------------------: | :--: | :--: | :--: | :--: | :--: | :--: |
>     |        Test error        | 0.88 | 0.71 | 0.49 | 0.44 | 0.32 | 0.23 |
>     | Algorithm DB variability | 0.78 | 0.68 | 0.50 | 0.49 | 0.37 | 0.21 |
>
>   - **Initial learning rate=0.01**
>
>     **(CIFAR-10)**
>
>     |      Training epoch      |  1   |  3   |  10  |  30  | 100  | 200  |
>     | :----------------------: | :--: | :--: | :--: | :--: | :--: | :--: |
>     |        Test error        | 0.43 | 0.25 | 0.16 | 0.11 | 0.06 | 0.06 |
>     | Algorithm DB variability | 0.41 | 0.23 | 0.15 | 0.11 | 0.05 | 0.05 |
>
>     **(CIFAR-100)**
>
>     |      Training epoch      |  1   |  3   |  10  |  30  | 100  | 200  |
>     | :----------------------: | :--: | :--: | :--: | :--: | :--: | :--: |
>     |        Test error        | 0.80 | 0.61 | 0.42 | 0.33 | 0.25 | 0.25 |
>     | Algorithm DB variability | 0.76 | 0.58 | 0.39 | 0.33 | 0.21 | 0.21 |
>
> From the tables, a clear positive correlation is observed between algorithm DB variability and test error, which fully supports the claims of our paper. Please kindly refer to Section 4.3 (Figure 2) in the revised manuscript for more details.
>
> **Q1.2:** *Label noise: if there exists some level of label noises and consequently epoch-wise double descent, does the DB variability still characterize the generalizability?*
>
> **A1.2:** Thanks. We conducted experiments following your suggestion, which fully support our arguments.
>
>
> We inject $20\\%$ label noise into CIFAR-10 and CIFAR-100. Then, $20$ ResNet-18 are trained on the noise data. Algorithm DB variability and test error are calculated at each epoch of the training process. Similarly, we draw (1) two curves of test error and algorithm DB variability curve with respect to the training time; and (2) a curve of algorithm DB variability to test error stops. The curves show a significant correlation between algorithm DB variability and test error.
>
> Please kindly refer the Section 4.4 (Figure 3(b)) in the revised manuscript for more details.

---

> > ### Comment · Reviewer_kiZ8 · 2021-11-20
> > **Thank you for your response**
> >
> > The experiments with label noise look convincing. In Figure 3 (b) Cifar-100 experiment, the variability curve seems to fall behind the test error curve, while in the other scenarios they are well aligned. Do the authors have thoughts on why this happens?
> >
> > Minor: You may add "with label noise" to Figure 3 (b) caption to distinguish it from Figure 2 (b).

---

> > > ### Author Response · Authors · 2021-11-21
> > > **Thanks for your question!**
> > >
> > > Thanks for this interesting question! Our conjecture is that the class number of data has a significant impact on the shape of the variability curves and test error curves. We would also love to note that the curves are still in full agreement with the negative correlation between the decision boundary variability and generalizability. It is very interesting to explore the mechanism behind the different behaviour of the curves when the class number is different. We endeavour to discover this mechanism in the future.
> > >
> > > To minor: Thanks. We will change the caption of Figure 3 (b) from "Algorithm DB variability vs. Epoch" to "Algorithm DB variability vs. Epoch (label noise)" in the revised manuscript.

---

> ### Comment · Reviewer_kiZ8 · 2021-11-20
> **raise rating to 6**
>
> I appreciate the efforts made by the authors on the additional experiments and revising the manuscript. The experiments further validify their claims, which together with the modifications resolve my questions.
>
> In general, in all cases I can think of, except for some cases where the authors give reasonable excuses, the decision boundary variability shows as a nice indicator of the model generalization, at least within a relative comparison among the tested models. Therefore, I would like to raise my score to 6. I'm still not convinced that the proposed metric can make reliable predictions on the generalization of arbitrary models because the gap between the metric and test error can be unpredictable, e.g. Fig. 3. For this reason, I don't further increase my score. Below, I respond to each reply separately.

---

> > ### Author Response · Authors · 2021-11-21
> > **Thanks!**
> >
> > Thank you very much for recognising our contributions!

---

### Official Review · Reviewer_3dWe · 2021-11-02

**Correctness:** 3
**Technical Novelty And Significance:** 3
**Empirical Novelty And Significance:** 3
**Recommendation:** 6
**Confidence:** 3

**Main Review:**

The proof and empirical results are sufficient to show the claim. All claims are clarified and clear to understand, the organization of the paper looks good.

(1) I have some issues concerning the theoretical aspects.

1. Assumptions 2--3 can be sound. These conditions are essential for the proof. When these conditions break down, what will we observe? In the binary classification, assumption 3 means the correct classification on average for any $(x, {\bf y}) \sim \mathcal{D}.$
2. Can you clarify the situation when the assumptions 2--3 are weakened?

(2) In addition, simulations studies to validate the claims are limited since we cannot know the causality between prediction variability and generalization due to various confounding effects. If you consider the causal graph to address this issue, the claim can be strong. In this case, I can only know the correlation, not causation.




**Summary Of The Paper:**

This paper studies that the smaller variability of prediction can provide a better generalization in empirical and theoretical foundations. Considered examples and theories are solid, and the intensive studies reveal the relationship (correlation not causation) between the variability of prediction and generalization.
The paper considers two types of variability with respect to algorithm and training data. In any case, lower variability ensures higher performance in generalization.

**Summary Of The Review:**

There is a valuable empirical analysis for the prediction variability. However, the more realistic cases should be considered.
Additionally, to validate the claim of the proposed, assumptions 2--3 should be checked in experiments.

---

> ### Author Response · Authors · 2021-11-18
> **To Reviewer 3dWe (2/2)**
>
> **Q2:** *In addition, simulations studies to validate the claims are limited since we cannot know the causality between prediction variability and generalization due to various confounding effects. If you consider the causal graph to address this issue, the claim can be strong. In this case, I can only know the correlation, not causation.*
>
> **A2:** Thanks for your suggestion. Intuitively, the well-trained network $f_\mathbb{Q}$ ($Q$) is affected by various effects ($Z$) such as network architectures, training strategies, and training data. On the other hand, $f_\mathbb{Q}$ determines the expected risk ($X=\mathcal{R}_\mathcal{D}(\mathbb{Q})$) and the decision boundary variability ($Y$). Thus, we draw a causal graph as follows
>
>   $$
>   Z\rightarrow Q\ ^{\nearrow^{\ \huge X}}_\{\searrow_\{\ \huge Y\}\}.
>   $$
>
>   From the casual graph, we observe that $X$ and $Y$ are both affected by $Q$. Hence, $X$ and $Y$ are connected by $Q$. Recalling that the significant correlation between $X$ and $Y$ observed in our experiments, the decision boundary variability $Y$ is also an excellent indicator of the generalizability $G$.
>
>   Moreover, we have conducted more experiments to explore the relationship between algorithm DB variability and training time, label noise, and training sample size, following your suggestion. The results are in full agreement with our claims.
>
> (1) **Training time:** we train ResNet-18 on CIFAR-10 and CIFAR-100 with different initial learning rate of $0.1$ and $0.01$. Based on the learned models, we calculate the algorithm DB variability and test error at each epoch of the training process as below:
>
>
>   - **Initial learning rate=0.1**
>
>     **(CIFAR-10)**
>
>     |      Training epoch      |  1   |  3   |  10  |  30  | 100  | 200  |
>     | :----------------------: | :--: | :--: | :--: | :--: | :--: | :--: |
>     |        Test error        | 0.58 | 0.41 | 0.22 | 0.19 | 0.09 | 0.05 |
>     | Algorithm DB variability | 0.50 | 0.39 | 0.23 | 0.22 | 0.12 | 0.05 |
>
>     **(CIFAR-100)**
>
>     |      Training epoch      |  1   |  3   |  10  |  30  | 100  | 200  |
>     | :----------------------: | :--: | :--: | :--: | :--: | :--: | :--: |
>     |        Test error        | 0.88 | 0.71 | 0.49 | 0.44 | 0.32 | 0.23 |
>     | Algorithm DB variability | 0.78 | 0.68 | 0.50 | 0.49 | 0.37 | 0.21 |
>
>   - **Initial learning rate=0.01**
>
>     **(CIFAR-10)**
>
>     |      Training epoch      |  1   |  3   |  10  |  30  | 100  | 200  |
>     | :----------------------: | :--: | :--: | :--: | :--: | :--: | :--: |
>     |        Test error        | 0.43 | 0.25 | 0.16 | 0.11 | 0.06 | 0.06 |
>     | Algorithm DB variability | 0.41 | 0.23 | 0.15 | 0.11 | 0.05 | 0.05 |
>
>     **(CIFAR-100)**
>
>     |      Training epoch      |  1   |  3   |  10  |  30  | 100  | 200  |
>     | :----------------------: | :--: | :--: | :--: | :--: | :--: | :--: |
>     |        Test error        | 0.80 | 0.61 | 0.42 | 0.33 | 0.25 | 0.25 |
>     | Algorithm DB variability | 0.76 | 0.58 | 0.39 | 0.33 | 0.21 | 0.21 |
>
>   From the tables, a clear positive correlation is observed between algorithm DB variability and test error, which fully supports the claims of our paper. Please kindly refer to Section 4.2 (Figure 2) in the revised manuscript for more details.
>
>   (2) **Label noise:** we inject $20\\%$ label noise into the training data and $20$ ResNet-18 are trained CIFAR-10 and CIFAR-100 with the noise labels. We then calculate algorithm DB variability and test error at each epoch of the training process. Similarly, we draw (1) two curves of test error and algorithm DB variability curve with respect to the training time; and (2) a curve of algorithm DB variability to test error stops. The curves show a significant correlation between algorithm DB variability and test error. Please kindly refer the Section 4.4 (Figure 3(b)) in the revised manuscript for more details.
>
>   (3) **Training sample size:** we train ResNet-18 with different sample size of $[2,000,\ 5,000,\ 10,000,\ 20,000,\ 50,000]$ on CIFAR-10 and CIFAR-100.  Based on these models, we calculate algorithm DB variability and test error as below:
>
>   - **CIFAR-10**
>
>     | Sample size | 2,000 | 5,000 | 10,000 | 20,000 | 50,000 |
>     | :---------: | :--: | :--: | :---: | :---: | :---: |
>     | Test error  | 0.44 | 0.27 | 0.15  | 0.09  | 0.05  |
>     | Algorithm DB variability| 0.33 | 0.19 | 0.11  | 0.07  | 0.05  |
>
>   - **CIFAR-100**
>
>     | Sample size | 2,000 | 5,000 | 10,000 | 20,000 | 50,000 |
>     | :---------: | :--: | :--: | :---: | :---: | :---: |
>     | Test error  | 0.82 | 0.67 | 0.48  | 0.33  | 0.22  |
>     | Algorithm DB variability| 0.57 | 0.48 | 0.36  | 0.26  | 0.21  |
>
>   From the tables, a clear positive correlation is observed between algorithm DB variability and test error, which fully supports the claims of our paper. Please kindly refer to Section 4.3 (Figure 3(a)) in the revised manuscript for more details.

---

> > ### Comment · Reviewer_3dWe · 2021-11-19
> > **Reply to Rebuttal**
> >
> > Thanks for your reply and intensive experiments. Your claim seems clear by using the causal graph. However, I cannot find a clear relationship between the proposed causal graph and experiments.
> > Finding correlation itself is valuable enough. Can it be true for all cases? It is a fundamental question. If it is true, then a more insightful explanation is required such that the regularity conditions hold for all cases. If not, it should be clarified when it is true or not. The regularity conditions of Theorems may tell about this issue.

---

> > > ### Author Response · Authors · 2021-11-19
> > > **Justifications for the causal graph**
> > >
> > > Thanks for your reply! We agree that the proposed causal graph is not delivered by the experiments; however, all the edges in the graph can be justified by sufficiently strong arguments to us.
> > >
> > > **$Z\rightarrow Q$:** $Z$ is defined as all the factors that would influence the learned networks, including network architecture, training strategy, sample size, etc. Thus, this edge is guaranteed.
> > >
> > > **$Q\rightarrow X$:** From the definition of expected risk $X$, it is determined by two factors, learned network $Q$ and loss function (which has been fixed in this case). Thus, the expected risk $X$ is determined by the learned network $Q$.
> > >
> > > **$Q\rightarrow Y$:** The decision boundary of the network $f_\boldsymbol{\theta}$ is defined to be the set $\\{\mathbf{x}\in \mathbb{R}^n| \exists i,j \in \{1,\cdots,k\}, i\neq j, f_\boldsymbol{\theta}^{(i)}(\mathbf{x})=f_\boldsymbol{\theta}^{(j)}(\mathbf{x}) =\max_q f^\{(q)\}_\boldsymbol{\theta}(\mathbf{x})\\}$,  where $\mathbf{x}$ is the input and $f_\boldsymbol{\theta}^{(i)}(\mathbf{x})$ denotes the $i$-th component of the output $f(\mathbf{x})$. Thus, the variability of decision boundary $Y$ is determined by the learned network $Q$.
> > >
> > > Moreover, in this paper, we only claim that we discover the “correlation” between the expected risk $X$ (or equivalently, the generalization) and the decision boundary variability $Y$. We agree that whether there is causality between them is a very interesting open problem. We will definitely work on this topic in the future.

---

> > > > ### Comment · Reviewer_3dWe · 2021-11-30
> > > > **Reply**
> > > >
> > > > Thanks for your reply. I want more insightful validation for the use of your work, related to Theorem. However, I think that your works are valuable enough.

---

> > > > > ### Author Response · Authors · 2021-11-30
> > > > > **Thanks!**
> > > > >
> > > > > Thank you very much for recognising our contribution and kind support!
> > > > >
> > > > > We will add more insightful validations following your suggestions.

---

> ### Author Response · Authors · 2021-11-18
> **To Reviewer 3dWe (1/2)**
>
> Thank you for your thorough review and constructive feedback. All your concerns have been carefully responded. The manuscript is carefully revised accordingly. We sincerely hope our responses fully address your questions.
>
> **Q1.1:** *Assumption 2--3 can be sound. These conditions are essential for the proof. When these conditions break down, what will we observe? Can you clarify the situation when the assumption 2--3 are weakened?*
>
> **A1.1:** Thanks for this interesting question! We have significantly extended our theory accordingly.
>
>  We first remove Assumptions 2-3. We prove that
> $$\text{Var}_\{(\mathbf{x},y)\sim \mathcal{D}\}\left[\mathbb{E}_\{\boldsymbol{\theta}\sim \mathbb{Q}\}\left[\mathbb{I}\left(y\notin T\left(f_\boldsymbol{\theta}, \mathbf{x}\right)\right)\right]\right] \leq \frac{AV(f_\mathbb{Q},\mathcal{D})+A+B}{k},$$
>
> where
>
> $A=2\sum_\{i>j\}\text{Cov}\left[\mathbb{E}_\{\boldsymbol{\theta}\sim \mathbb{Q}\}\left[\mathbb{I}\left(T\left(f_\boldsymbol{\theta}, \mathbf{x}\right)=i\right)\mathbb{I}\left(y\notin T\left(f_\boldsymbol{\theta}, \mathbf{x}\right) \right)\right], \mathbb{E}_\{\boldsymbol{\theta}\sim \mathbb{Q}\}\left[\mathbb{I}\left(T\left(f_\boldsymbol{\theta}, \mathbf{x}\right)=j\right)\mathbb{I}\left(y\notin T\left(f_\boldsymbol{\theta}, \mathbf{x}\right)\right)\right]\right]$
>
> and
>
>  $B=(k+1)\mathbb{E}_\{(\mathbf{x},y)\sim\mathcal{D}\}\mathbb{E}^2_\{\boldsymbol{\theta}\sim \mathbb{Q}\}\left[\mathbb{I}\left(T\left(f_\boldsymbol{\theta}, \mathbf{x}\right)=i\right)\mathbb{I}\left(y\notin T\left(f_\boldsymbol{\theta}, \mathbf{x}\right)\right)\right]-2\mathbb{E}_\{(\mathbf{x},y)\sim\mathcal{D}\}\mathbb{E}_\{\boldsymbol{\theta}\sim \mathbb{Q}\}\left[\mathbb{I}\left(y\notin T\left(f_\boldsymbol{\theta}, \mathbf{x}\right)\right)\right]$
>
>
> Compared with the original version,
> $$\text{Var}_\{(\mathbf{x},y)\sim \mathcal{D}\}\left[\mathbb{E}_\{\boldsymbol{\theta}\sim \mathbb{Q}\}\left[\mathbb{I}\left(y\notin T\left(f_\boldsymbol{\theta}, \mathbf{x}\right)\right)\right]\right] \leq \frac{AV(f_\mathbb{Q},\mathcal{D})}{k}.$$
>
>   Assumptions 2-3 ensure that $A\leq0$ and $B\leq0$, respectively, so that we may remove them for brevity.
>
>   We have also relaxed Assumption 2 and 3 as below,
>
>   $$
>   \mathbb{E}_\{(\mathbf{x},y)\sim \mathcal{D}\}\mathbb{E}^2_\{\boldsymbol{\theta}\sim \mathbb{Q}\}\left[\mathbb{I}\left(y\notin T\left(f_\boldsymbol{\theta}, \mathbf{x}\right)\right)\right] \leq \frac{2}{k+1}\mathcal{R}_\{\mathcal{D}\}(\mathbb{Q}),
>   $$
>
>   Using this new assumption, we prove the following new version of Lemma 1:
>
>   $$
>   \text{Var}_\{(\mathbf{x},y)\sim \mathcal{D}\}\left[\mathbb{E}_\{\boldsymbol{\theta}\sim \mathbb{Q}\}\left[\mathbb{I}\left(y\notin T\left(f_\boldsymbol{\theta}, \mathbf{x}\right)\right)\right]\right] \leq \frac{AV(f_\mathbb{Q},\mathcal{D})}{k-1}.
>   $$
>
>   Please kindly refer to Section 4.5.2 in the revised manuscript for more details.
>
> **Q1.2:** *In the binary classification, assumption 3 means the correct classification on average for any $(\mathbf{x},y)\sim \mathcal{D}$*
>
>   **A1.2:** Yes, we agree.

---

### Official Review · Reviewer_pU8o · 2021-11-02

**Correctness:** 4
**Technical Novelty And Significance:** 3
**Empirical Novelty And Significance:** 2
**Recommendation:** 6
**Confidence:** 3

**Main Review:**


Overall, I think the paper introduces a number of interesting potential ideas for explaining generalization in neural networks.  But my initial read leads me to a number of questions and concerns before I can more confidently back this paper for acceptance; I've listed a number of them below.


(1) For the central 'algorithm DB variability' claims, the primary experimental validation relies upon the three algorithmic variations of "standard training w/ data augmentation", "adversarial training", and "standard training without data augmentation".  My understanding is the primary purpose of these set-ups is to produce a plot like in Figure 2(c), comparing generalization of fixed models but changing algorithmic procedures, so as to discern a relationship between generalization and DB variability.  I feel the strength of this message would be improved if a small number of additional experiments further tested this hypothesis.  Some possible ideas: using different early-stopping schedules, or using progressively larger data subsets (e.g. using 10,000, 20,000, 30,000 training samples).  This would allow for a fixed model but a change in other parameters that can affect test accuracy.

(2) The usage of BigGAN images to estimate variability seems tolerable but it would be best if there were some additional confirmation that the obvious alternative -- data subsetting, say keeping 5,000 random samples from the training set as a new test set and not using these in the training process -- would result in similar trends.


(3) What is the intuition for why Assumption 3 is needed?  Can the authors give a verbal description of how this places a role in the proof in the main section?


Some minor points/typos etc.:

p.1 and a few places: the notion of "margin" seems a bit muddled throughout.  The margin of a classifier f(x) is the largest distance gamma s.t. y * f(x) >= gamma.  The authors seem to be referring to the margin at a point x in terms of the largest rho s.t. y * f(x') has the same sign as y * f(x) for all ||x-x'|| <= rho; these are closely related only if f has a small Lipschitz constant as a function of x.  It is definitely the case that large margins in the former sense have good generalization performance (vs. authors' suggestion in p.1); this is shown in standard statistical learning theory.  But adversarial robustness  (the latter sense of margin) does not imply good generalization.


p.4, Sec. 4.1: Soudry et al. (2018) isn't really about neural networks; better references would be Lyu & Li (2020) and Ji & Telgarsky (2020) not 2018a ("directional convergence and alignment").

p.4: in Fig. 1, the authors say (c) plots DB variability as a function of sample complexity, but x axis is nonlinearity level?

p.7: Assumption 2 is missing an "=j" inside the second expectation

Appendix A.1: what adversarial training method is used?

Appendix B.2: how does Def B.1 differ from eta, epsilon DB variability?

Appendix C: the E^2() notation is weird; does this mean [E()]^2 or E[()^2]?
How does one go from (50)-(51)?   Not clear on first read.


**Summary Of The Paper:**

The authors introduce the notion of decision boundary (DB) variability and develop generalization bounds in terms of this.  They consider two notions, that of 'algorithm DB variability' and 'epsilon-eta DB variability', the latter a generalization of the former.  They show upper and lower bounds for the risk of classifiers in terms of DB variability under a variety of assumptions, as well as experimentally verify some of their claims.

**Summary Of The Review:**

Overall, I think the paper introduces a number of interesting potential ideas for explaining generalization in neural networks.  But my initial read leads me to a number of questions and concerns before I can more confidently back this paper for acceptance.

---

> ### Author Response · Authors · 2021-11-17
> **To Reviewer pU8o (3/3)**
>
> **To minor points/typos:**
>
> **Q4:** *p.1 and a few places: the notion of "margin" seems a bit muddled throughout. The margin of a classifier f(x) is the largest distance gamma s.t. y\*f(x) >= gamma. The authors seem to be referring to the margin at a point x in terms of the largest rho s.t. y\*f(x') has the same sign as y\*f(x) for all ||x-x'|| <= rho; these are closely related only if f has a small Lipschitz constant as a function of x. It is definitely the case that large margins in the former sense have good generalization performance (vs. authors' suggestion in p.1); this is shown in standard statistical learning theory. But adversarial robustness (the latter sense of margin) does not imply good generalization.*
>
>   **A4:** Thanks for your suggestion. We have carefully revised our manuscript to avoid using the term "margin". The related statements have been removed.
>
> **Q5:** *p.4, Sec. 4.1: Soudry et al. (2018) isn't really about neural networks; better references would be Lyu & Li (2020) and Ji & Telgarsky (2020) not 2018a ("directional convergence and alignment").*
>
>  **A5:** Thanks and addressed.
>
>  **Q6:** *p.4: in Fig. 1, the authors say (c) plots DB variability as a function of sample complexity, but x axis is nonlinearity level?*
>
>   **A6:** Thanks. We have carefully revised our presentation to make it clear. The “sample complexity” and “non-linearity level” have been removed.
>
> **Q7:** *p.7: Assumption 2 is missing an "=j" inside the second expectation*
>
>   **A7:** Thanks and addressed.
>
>  **Q8:** *Appendix A.1: what adversarial training method is used?*
>
>   **A8:** Thanks. Projected gradient descent (PGD) [1] is adopted in our experiments, as shown in Appendix A.3. Implementation details are as below.
>
>
>   For the experiments in Section 4.1, the adversarial radius is set as $10/255$ and $l_\infty$ distance is chosen for adversarial training.
>
>   All the hyperparameters are selected via grid search and cross-validation.
>
> **Q9:** *Appendix B.2: how does Def B.1 differ from eta, epsilon DB variability?*
>
>   **A9:** Thanks. They are the same. We have carefully revised our manuscript to avoid any ambiguity.
>
>  **Q10.1:** *Appendix C: the E^2() notation is weird; does this mean [E()]^2 or E[()^2]?*
>
>   **A10.1:** Thanks. $\mathbb{E}^2()$ means $[\mathbb{E}()]^2$. We will add an explanation in the revised manuscript.
>
> **Q10.2:** *How does one go from (50)-(51)? Not clear on first read.*
>
>   **A10.2:** Thanks. The procedure is presented below,
>
> From Assumption 3, we have that for $\forall (\mathbf{x},y)\sim \mathcal{D}$ and $i\in [k]$,
> $$\mathbb{E}_{\boldsymbol{\theta}\sim \mathbb{Q}}\left[\mathbb{I}\left(T\left(f_\boldsymbol{\theta}, \mathbf{x}\right)=i\right)\mathbb{I}\left(y\notin T\left(f_\boldsymbol{\theta}, \mathbf{x}\right)\right)\right] \leq \frac{2}{k+2}.$$
>
> Recall that $y=1$ for all examples, we have that
>  $$
>  \sum_{k=2}^k \mathbb{E}^2_{\boldsymbol{\theta}\sim \mathbb{Q}}\left[\mathbb{I}\left(T\left(f_\boldsymbol{\theta}, \mathbf{x}\right)=i\right)\right] \leq  \frac{2}{k+2} \sum_{k=2}^k \mathbb{E}_{\boldsymbol{\theta}\sim \mathbb{Q}}\left[\mathbb{I}\left(T\left(f_\boldsymbol{\theta}, \mathbf{x}\right)=i\right)\right].
>  $$
>
> Then, we calculate the expectation over data generation distribution $\mathcal{D}$ on both sides, and get the following inequality,
>   $$
>   \sum_{i=2}^k \mathbb{E}_\{(\mathbf{x},y)\sim \mathcal{D}\}\mathbb{E}_\{\boldsymbol{\theta}\sim \mathbb{Q}\}^2\left[\mathbb{I}\left(T\left(f_\boldsymbol{\theta}, \mathbf{x}\right) = i\right)\right] - \sum_\{i=2\}^k \mathbb{E}_\{(\mathbf{x},y)\sim \mathcal{D}\} \mathbb{E}_\{\boldsymbol{\theta}\sim \mathbb{Q}\}\left[\mathbb{I}\left(T\left(f_\boldsymbol{\theta}, \mathbf{x}\right) = i\right)\right] \\
>  \leq - \frac{k}{2} \sum_\{i=2\}^k \mathbb{E}_\{(\mathbf{x},y)\sim \mathcal{D}\}\mathbb{E}_\{\boldsymbol{\theta}\sim \mathbb{Q}\}^2\left[\mathbb{I}\left(T\left(f_\boldsymbol{\theta}, \mathbf{x}\right) = i\right)\right].
>   $$
>
>
> In this way, we go from eq. (50) to eq. (51).
>
>   We have carefully refined our proof in the manuscript to improve the readability (Assumptions 2 and 3 have been relaxed following your suggestions).
>
>   [1] A. Madry, A. Makelov, L. Schmidt, D. Tsipras, and A. Vladu. “Towards deep learning models resistant to adversarial attacks.” In *International Conference on Learning Representations*, 2018.

---

> > ### Comment · Reviewer_pU8o · 2021-11-30
> > **thanks**
> >
> > Thanks to the authors for the very extensive additional experiments.  I think these additional experiments make it a more convincing work about generalization in neural network training.  I think the correlation between the variability metric and generalization performance is quite impressive, and the curious behavior in the presence of label noise is interesting.  Although the general trend for DB variability and test accuracy seem aligned, it seems there is a large gap between the test accuracy and the DB variability; I'm wondering if this holds for other datasets and NN models.  I would recommend the authors further investigate this.
> >
> > I'm increasing my score to a 6.  The reason for not increasing my score further is that, although the paper's does make some interesting contributions, many of these were made after the initial submission, and I think the overall story of the paper could be improved given the extensive new experiments.  For instance, it might make more sense to focus on the early stopping / data subsetting / label noise angles rather than the adversarial training angle.  I would also be sure to emphasize the investigatory nature of the work, and to lean less on claims that might be construed as "causal" of generalization.  I believe if these improvements are made, the paper would be a strong one.

---

> > > ### Author Response · Authors · 2021-11-30
> > > **Thanks!**
> > >
> > > Thank you very much for recognising our contributions and kind support!
> > >
> > > Your concerns have been responded below. The manuscript will be carefully revised accordingly in the next version.
> > >
> > > **Q1:** *Although the general trend for DB variability and test accuracy seem aligned, it seems there is a large gap between the test accuracy and the DB variability; I'm wondering if this holds for other datasets and NN models. I would recommend the authors further investigate this.*
> > >
> > > **A1:** Thanks. We would love to clarify that we do not intend to claim the alignment of the curves of DB variability and test error; we only hope to report the positive correlation between them; please kindly refer to Fig. 2(b). We will carefully revise our manuscript to avoid any ambiguity. Moreover, we will follow your recommendation to further investigate this correlation.
> > >
> > > **Q2:** _I think the overall story of the paper could be improved given the extensive new experiments. For instance, it might make more sense to focus on the early stopping / data subsetting / label noise angles rather than the adversarial training angle. I would also be sure to emphasize the investigatory nature of the work, and to lean less on claims that might be construed as "causal" of generalization. I believe if these improvements are made, the paper would be a strong one._
> > >
> > > **A2:** Thank you very much for your valuable suggestions! We will further revise our manuscript following your suggestions. The presentation will be revised from the adversarial training angle to the early stopping/data subsetting/label noise angles. We will also emphasize the investigatory nature of this work, and tune down our claims that might be construed as "causal" of generalization.

---

> ### Author Response · Authors · 2021-11-17
> **To Reviewer pU8o (2/3)**
>
> **Q2:** *The usage of BigGAN images to estimate variability seems tolerable but it would be best if there were some additional confirmation that the obvious alternative -- data subsetting, say keeping 5,000 random samples from the training set as a new test set and not using these in the training process -- would result in similar trends.*
>
>  **A2:** Thanks. We have conducted experiments following your suggestion, which fully support our claims.
>
> We randomly divide the test set of $10,000$ examples into two parts of equal size, one for estimating algorithm DB variability, and the other for estimating test error. ResNet-18 is then trained on CIFAR-10 and CIFAR-100 with different training strategies (standard, adversarial training, and without data augmentation) and sample sizes of $[2,000,\ 5,000,\ 10,000,\ 20,000,\ 50,000]$. The results are presented as follows,
>
>   - **CIFAR-10**
>
>     |    Training strategy     | Standard | Without data augmentation | Adversarial |
>     | :----------------------: | :------: | :-----------------------: | :---------: |
>     |        Test error        | 0.04821  |          0.1213           |   0.1806    |
>     | Algorithm DB variability | 0.04526  |          0.1155           |   0.1275    |
>
>     |   Training sample size   |  2,000  |  5,000  | 10,000  | 20,000  | 50,000  |
>     | :----------------------: | :----: | :----: | :----: | :----: | :----: |
>     |        Test error        | 0.4410 | 0.2709 | 0.1499 | 0.0862 | 0.0482 |
>     | Algorithm DB variability | 0.3572 | 0.2117 | 0.1260 | 0.0774 | 0.0447 |
>
>   - **CIFAR-100**
>
>     |    Training strategy     | Standard | Without data augmentation | Adversarial |
>     | :----------------------: | :------: | :-----------------------: | :---------: |
>     |        Test error        |  0.2200  |          0.3810           |   0.4620    |
>     | Algorithm DB variability |  0.1887  |          0.3746           |   0.3802    |
>
>     |   Training sample size   |  2,000  |  5,000  | 10,000  | 20,000  | 50,000  |
>     | :----------------------: | :----: | :----: | :----: | :----: | :----: |
>     |        Test error        | 0.8200 | 0.6636 | 0.4792 | 0.3281 | 0.2205 |
>     | Algorithm DB variability | 0.5714 | 0.4900 | 0.3821 | 0.2763 | 0.1887 |
>
> From the tables, a clear positive correlation is observed between algorithm DB variability and test error in all scenarios, which fully supports the claims of our paper.
>
> We would also like to justify our motivation for using BigGAN to generate images. Using BigGAN, we can generate arbitrarily large dataset, which significantly facilitates our empirical study. In our experiments, $200,000$ images are generated. Please kindly refer to Appendix A.1 in the revised manuscript for more details.
>
> **Q3:** *What is the intuition for why Assumption 3 is needed? Can the authors give a verbal description of how this places a role in the proof in the main section?*
>
>  **A3:**  Thanks for this interesting question!
>
> Assumption 3 is introduced to control the variance of class-wise risk $$\mathbb{E}_{\boldsymbol{\theta}\sim \mathbb{Q}}[\mathbb{I}\left( T\left(f_\boldsymbol{\theta}, \mathbf{x}\right)=i\right)\mathbb{I}\left(y\notin T\left(f_\boldsymbol{\theta}, \mathbf{x}\right)\right)],$$
>
> so that the variance of the total risk $\mathbb{E}_{\boldsymbol{\theta}\sim \mathbb{Q}}[\mathbb{I}\left(y\notin T\left(f_\boldsymbol{\theta}, \mathbf{x}\right)\right)]$ is bounded using algorithm DB variability.
>
> Assumption 3 can be removed without loss of significance of our theory. One of our main result,
> $$\text{Var}_{(\mathbf{x},y)\sim \mathcal{D}}\left[\mathbb{E}_\{\boldsymbol{\theta}\sim \mathbb{Q}\}\left[\mathbb{I}\left(y\notin T\left(f_\boldsymbol{\theta}, \mathbf{x}\right)\right)\right]\right] \leq \frac{AV(f_\mathbb{Q},\mathcal{D})}{k},$$
> becomes
> $$\text{Var}_\{(\mathbf{x},y)\sim \mathcal{D}\}\left[\mathbb{E}_\{\boldsymbol{\theta}\sim \mathbb{Q}\}\left[\mathbb{I}\left(y\notin T\left(f_\boldsymbol{\theta}, \mathbf{x}\right)\right)\right]\right] \leq \frac{AV(f_\mathbb{Q},\mathcal{D})+B}{k}.$$
>
> Assumption 3 helps remove the term $B$ for brevity. All other parts of the proofs only need very minor revisions.
>
>  We have also relaxed Assumptions 2 and 3, as the following “new Assumption 2,"
>   $$
>   \mathbb{E}_\{(\mathbf{x},y)\sim \mathcal{D}\}\mathbb{E}^2_\{\boldsymbol{\theta}\sim \mathbb{Q}\}\left[\mathbb{I}\left(y\notin T\left(f_\boldsymbol{\theta}, \mathbf{x}\right)\right)\right] \leq \frac{2}{k+1}\mathcal{R}_\mathcal{D}(\mathbb{Q}).
>   $$
>
>  Under this new assumption, we can still prove the Lemma 1 in the manuscript:
> $$\text{Var}_{(\mathbf{x},y)\sim \mathcal{D}}\left[\mathbb{E}_\{\boldsymbol{\theta}\sim \mathbb{Q}\}\left[\mathbb{I}\left(y\notin T\left(f_\boldsymbol{\theta}, \mathbf{x}\right)\right)\right]\right] \leq \frac{AV(f_\mathbb{Q},\mathcal{D})}{k-1}$$
>
> All other parts of the original proofs do not need to revise.
>
> Please also kindly refer to Section 4.5.2 in the revised manuscript for more details.

---

> ### Author Response · Authors · 2021-11-17
> **To Reviewer pU8o (1/3)**
>
> Thank you very much for your thorough review and constructive feedback. All your concerns have been carefully responded. The manuscript is carefully revised accordingly. We sincerely hope our responses fully address your questions.
>
> **To major concerns:**
>
> **Q1:** *I feel the strength of this message would be improved if a small number of additional experiments further tested this hypothesis. Some possible ideas: using different early-stopping schedules, or using progressively larger data subsets (e.g. using 10,000, 20,000, 30,000 training samples).*
>
> **A1:** Thanks. We have conducted experiments following your suggestion, which fully support our claims.
>
>   (1) **Early-stopping:** we train ResNet-18 on CIFAR-10 and CIFAR-100 with initial learning rates of $0.1$ and $0.01$.  Based on the learned models, we calculate the algorithm DB variability and test error at each epoch of the training process as below,
>
>   - **Initial learning rate=0.1**
>
>     **(CIFAR-10)**
>
>     |      Training epoch      |  1   |  3   |  10  |  30  | 100  | 200  |
>     | :----------------------: | :--: | :--: | :--: | :--: | :--: | :--: |
>     |        Test error        | 0.58 | 0.41 | 0.22 | 0.19 | 0.09 | 0.05 |
>     | Algorithm DB variability | 0.50 | 0.39 | 0.23 | 0.22 | 0.12 | 0.05 |
>
>     **(CIFAR-100)**
>
>     |      Training epoch      |  1   |  3   |  10  |  30  | 100  | 200  |
>     | :----------------------: | :--: | :--: | :--: | :--: | :--: | :--: |
>     |        Test error        | 0.88 | 0.71 | 0.49 | 0.44 | 0.32 | 0.23 |
>     | Algorithm DB variability | 0.78 | 0.68 | 0.50 | 0.49 | 0.37 | 0.21 |
>
>   - **Initial learning rate=0.01**
>
>     **(CIFAR-10)**
>
>     |      Training epoch      |  1   |  3   |  10  |  30  | 100  | 200  |
>     | :----------------------: | :--: | :--: | :--: | :--: | :--: | :--: |
>     |        Test error        | 0.43 | 0.25 | 0.16 | 0.11 | 0.06 | 0.06 |
>     | Algorithm DB variability | 0.41 | 0.23 | 0.15 | 0.11 | 0.05 | 0.05 |
>
>     **(CIFAR-100)**
>
>     |      Training epoch      |  1   |  3   |  10  |  30  | 100  | 200  |
>     | :----------------------: | :--: | :--: | :--: | :--: | :--: | :--: |
>     |        Test error        | 0.80 | 0.61 | 0.42 | 0.33 | 0.25 | 0.25 |
>     | Algorithm DB variability | 0.76 | 0.58 | 0.39 | 0.33 | 0.21 | 0.21 |
>
>   From the tables, a clear positive correlation is observed between algorithm DB variability and test error, which fully supports the claims of our paper. Please kindly refer to Section 4.2 (Figure 2) in the revised manuscript for more details.
>
>   (2) **Data subsets:** we train ResNet-18 with different sample size of $[2,000,\ 5,000,\ 10,000,\ 20,000,\ 50,000]$ on CIFAR-10 and CIFAR-100.  Based on the learned models, we calculate algorithm DB variability and test error as below,
>
>   - **CIFAR-10**
>
>     |       Sample size        | 2,000 | 5,000 | 10,000 | 20,000 | 50,000 |
>     | :----------------------: | :---: | :---: | :----: | :----: | :----: |
>     |        Test error        | 0.44  | 0.27  |  0.15  |  0.09  |  0.05  |
>     | Algorithm DB variability | 0.33  | 0.19  |  0.11  |  0.07  |  0.05  |
>
>   - **CIFAR-100**
>
>     |       Sample size        | 2,000 | 5,000 | 10,000 | 20,000 | 50,000 |
>     | :----------------------: | :---: | :---: | :----: | :----: | :----: |
>     |        Test error        | 0.82  | 0.67  |  0.48  |  0.33  |  0.22  |
>     | Algorithm DB variability | 0.57  | 0.48  |  0.36  |  0.26  |  0.21  |
>
>   From the tables, a clear positive correlation is observed between algorithm DB variability and test error, which fully supports the claims of our paper. Please kindly refer to Section 4.3 (Figure 3(a)) in the revised manuscript for more details.

---

### Official Review · Reviewer_fXBi · 2021-11-02

**Correctness:** 4
**Technical Novelty And Significance:** 4
**Empirical Novelty And Significance:** 4
**Recommendation:** 8
**Confidence:** 3

**Main Review:**

The notions introduced are rigorous and prove useful. The theoretical claims are supported by some empirical work as well. The bounds do not depend on network size and apparently reflect the behavior of practical and popular networks.

Several bounds are presented, and while it is stated that they do not depend on model size -- implying other existing bounds do -- there is no comparison presented. When would these bound be more useful than existing ones? When are other be more appropriate (or tighter)? If these questions seem misguided, perhaps a short discussion of why the bounds derived do not need to be compared to existing ones would be helpful.

I admit I am not so familiar with this domain and may have overlooked something.

**Summary Of The Paper:**

Decision boundary variability is measured in two ways. One method depends on the algorithm, or the random seed and reflects how much the boundaries change wen retraining the same network on the same data. The other technique reflects the variability across different amounts of training data. Generalization bounds derived from these two quantities are presented.

**Summary Of The Review:**

The derivations of new bounds that utilize the proposed decision boundary variability measures are strong. The claims are supported by experimental results where appropriate. This paper would be even stronger with more detail/discussion about existing work.

---

> ### Author Response · Authors · 2021-11-16
> **To Reviewer fXBi**
>
> Thank you very much for your constructive comments and kind support! All your concerns have been carefully responded. The manuscript has been revised accordingly.
>
> **Q1:** *There is no comparison between upper bounds. When would these bound be more useful than existing ones? When are other be more appropriate (or tighter)*
>
>   **A1:**  Thanks. The discussion and comparison about two upper bounds, given in Theorem 3 and Theorem 4, are presented as follows.
>
> The bounds are obtained from algorithm DB variability and data DB variability, respectively. They are of orders
> $\mathcal{O}\left(\sqrt{AV(f_{\mathbb{Q}}, \mathcal{D})/m(k-1)}\right)$
> and
> $\mathcal{O}\left(\frac{1}{\sqrt{m}}+\epsilon+\eta\log\frac{1}{\eta}\right),$ where $m$ is the sample size, $k$ is the number of classes, $AV(f_{\mathbb{Q}}, \mathcal{D})$ is the algorithm DB variability, and $(\epsilon,\eta)$ are parameters of data DB variability. Detailed comparisons between them are given below:
>
> 1.	Theorem 3 and Theorem 4 characterize the impact from algorithm and data on the generalizability, respectively;
> 2.	Compared with Theorem 4, Theorem 3 relies on additional Assumptions 2 and 3 (which have been relaxed to a new weaker Assumption 2; please kindly refer to A3 to Reviewer pU8o, A1.1 to Reviewer 3dWe, and Section 4.5.2 in the revised manuscript for more details);
> 3.	Theorem 3 can better characterize the impact of the class number $k$ on the generalizability;
> 4.	Theorem 3 can be better empirically calculated: algorithm DB variability $AV(f_{\mathbb{Q}}, \mathcal{D})$ is easier to calculate than data DB variability $(\epsilon,\eta)$ in practice.
>
>   Many existing generalization bounds based on hypothesis complexity, such as VC dimension [1], Rademacher complexity [2, 3], and cover number [2, 3], require access to the network weight. In contrast, our generalization bounds only require access to the network predictions. Hence, our bounds have advantages in empirically approximating the generalization bound in (1) black-model settings, where model parameters are unavailable; and (2) over-parameterized settings, where calculating the weight norm is of prohibitively high computing burden.
>
>   [1] P. L. Bartlett, N. Harvey, C. Liaw, and A. Mehrabian. “Nearly tight VC-dimension and pseudodimension bounds for piecewise linear neural networks.” *Journal of Machine Learning Research*, 20(63):1–17, 2019.
>
>   [2] P. L. Bartlett, D. J. Foster, and M. Telgarsky. "Spectrally-normalized margin bounds for neural networks." In *Advances in Neural Information Processing Systems*, 2017.
>
>   [3] N. Golowich, A. Rakhlin, and O. Shamir. "Size-independent sample complexity of neural networks." In *Conference On Learning Theory*, 2018.

---

> > ### Comment · Reviewer_fXBi · 2021-11-21
> > **Thanks for the response**
> >
> > Thank you for thoroughly addressing my comments. I will keep my score as is.

---

> > > ### Author Response · Authors · 2021-11-30
> > > **Thanks!**
> > >
> > > Thank you very much for your kind support!

---

### Decision · Program_Chairs · 2022-01-20

**Decision:**

Reject

**Comment:**

This paper proposes the notation of DB variability, which is essentially prediction variance. It is also closely related to algorithmic stability which is a theoretically more sound notation to derive generalization bounds. The paper is a mixed bag of empirical observations and "theory". However, looking at the "theoretical results" in the paper, it is clear that the authors lack adequate theoretical background.

I'd be more positive if the paper has been focused more on the former, and could be judged by the empirical part only. While the reviews were positive, I looked at them and realized that similar to the authors, the reviewers also lack theoretical backgrounds.

First, the fact large variance implies a generalization lower bound is trivial, to the degree it is not worth stating as a "result". Second small variance implies a generalization upper bound isn't true. One can have a predictor that perfectly overfits the training data and predicts class 0 everywhere else. This has small prediction variance but poor generalization. In this context, the upper bound analysis of the paper is clearly misleading. Usually one compares training error to generalization error, where the estimator depends on the training set. In such case, one cannot use the simple argument of the convergence of the empirical mean of sum of independent random variables to the mean due to the dependency of estimator on the training set, e.g. in Thm 3. One needs to use uniform convergence and exponential probability (instead of Chebyshev) inequality to obtain such results. The right hand side of Thm 3 (the theorem itself is also very poorly stated. and shouldn't be allowed to be published) could not be interpreted as training error as should usually be the case for such bounds, but only as validation error. Such a result (comparing validation and test error when distribution isn't changed) has no value.

I would not elaborate on other similar issues.  My recommendation is to focus on the empirical study if the authors are not familiar with theoretical analysis.